# Integrative analysis of scRNA-seq and scATAC-seq revealed transit-amplifying thymic epithelial cells expressing autoimmune regulator

Takahisa Miyao[1,2], Maki Miyauchi[1,2], S Thomas Kelly[3], Tommy W Terooatea[3], Tatsuya Ishikawa[1,2], Eugene Oh[1], Sotaro Hirai[1], Kenta Horie[1], Yuki Takakura[1], Houko Ohki[1,2], Mio Hayama[1,2], Yuya Maruyama[1,2], Takao Seki[1], Hiroto Ishii[1,2], Haruka Yabukami[3], Masaki Yoshida[4], Azusa Inoue[5], Asako Sakaue-Sawano[6], Atsushi Miyawaki[6], Masafumi Muratani[7], Aki Minoda[3], Nobuko Akiyama[1]*, Taishin Akiyama[1,2]*

[1]Laboratory for Immune Homeostasis, RIKEN Center for Integrative Medical Sciences, Yokohama, Japan; [2]Immunobiology, Graduate School of Medical Life Science, Yokohama City University, Yokohama, Japan; [3]Laboratory for Cellular Epigenomics, RIKEN Center for Integrative Medical Sciences, Yokohama, Japan; [4]YCI Laboratory for Immunological Transcriptomics, RIKEN Center for Integrative Medical Sciences, Kanagawa, Japan; [5]YCI Laboratory for Metabolic Epigenetics, RIKEN Center for Integrative Medical Sciences, Kanagawa, Japan; [6]Laboratory for Cell Function Dynamics, RIKEN Center for Brain Science, Saitama, Japan; [7]Transborder Medical Research Center, and Department of Genome Biology, Faculty of Medicine, University of Tsukuba, Ibaraki, Japan

*For correspondence:
nobuko.akiyama@riken.jp (NA);
taishin.akiyama@riken.jp (TA)

Competing interest: The authors declare that no competing interests exist.

**Abstract** Medullary thymic epithelial cells (mTECs) are critical for self-tolerance induction in T cells via promiscuous expression of tissue-specific antigens (TSAs), which are controlled by the transcriptional regulator, AIRE. Whereas AIRE-expressing (Aire+) mTECs undergo constant turnover in the adult thymus, mechanisms underlying differentiation of postnatal mTECs remain to be discovered. Integrative analysis of single-cell assays for transposase-accessible chromatin (scATAC-seq) and single-cell RNA sequencing (scRNA-seq) suggested the presence of proliferating mTECs with a specific chromatin structure, which express high levels of Aire and co-stimulatory molecules, CD80 (Aire+CD80hi). Proliferating Aire+CD80hi mTECs detected using Fucci technology express a minimal number of Aire-dependent TSAs and are converted into quiescent Aire+CD80hi mTECs expressing high levels of TSAs after a transit amplification. These data provide evidence for the existence of transit-amplifying Aire+mTEC precursors during the Aire+mTEC differentiation process of the postnatal thymus.

## Editor's evaluation

This report shows by scRNAs-seq and scATAC-seq the presence of a population of proliferating medullary thymic epithelial cells (mTECs) with a specific chromatin structure and high expression of Aire and CD80. Such Aire-expressing transit-amplifying mTECs may play a key role in establishing immunological self-tolerance.

## Introduction

Medullary thymic epithelial cells (mTECs) are essential for induction of T cell self-tolerance in the thymus (*Abramson and Anderson, 2017*; *Inglesfield et al., 2019*). mTECs ectopically express thousands of tissue-specific antigens (TSAs), and this expression is regulated by transcription factors, AIRE and FEZF2 (*Anderson et al., 2002*, *Takaba et al., 2015*). TSAs are directly or indirectly presented to developing T cells, and T cells that recognize TSAs with high affinity undergo apoptosis or are converted into regulatory T cells, thereby suppressing the onset of autoimmune diseases (*Abramson and Anderson, 2017*; *Inglesfield et al., 2019*).

Several studies have suggested processes and underlying mechanisms of mTEC differentiation during thymic organogenesis (*Abramson and Anderson, 2017*; *Inglesfield et al., 2019*; *Rossi et al., 2007*; *Akiyama et al., 2016*; *Akiyama et al., 2005*; *Akiyama et al., 2008*; *Hikosaka et al., 2008*; *Mouri et al., 2011*; *Kajiura et al., 2004*). In addition, some previous studies have reported that mTEC turnover is homeostatic in the adult thymus, with a duration of approximately 2 weeks (*Gäbler et al., 2007*; *Gray et al., 2007*; *Gray et al., 2006*). Notably, however, cellular mechanisms underlying maintenance of adult mTECs remain unclear. mTEC subpopulations are largely classified based on their expression of cell surface markers (mainly CD80 and MHC class II) and Aire in the adult thymus (*Abramson and Anderson, 2017*). CD80$^{lo}$ and Aire-negative (Aire$^-$) mTECs (mTEC$^{lo}$) are thought to be immature, and they differentiate into CD80$^{hi}$ Aire-expressing (Aire$^+$) mTECs that are reportedly post-mitotic (*Gray et al., 2007*). Aire$^+$ mTECs are further converted into Aire-negative mTECs (post-Aire mTECs) (*Metzger et al., 2013*; *Michel et al., 2017*; *Nishikawa et al., 2014*; *Wang et al., 2012*; *White et al., 2010*). Moreover, a previous study suggested that mTECs might be differentiated from stage-specific embryonic antigen-1$^+$ (SSEA-1) claudine3/4$^+$ mTEC stem cells (*Sekai et al., 2014*). These views are primarily based on fate mapping studies involving transfer and reaggregation of sorted cell populations with fetal thymus (*Rossi et al., 2007*; *Gray et al., 2007*; *Sekai et al., 2014*) and on experiments employing genetic marking (*Metzger et al., 2013*; *Nishikawa et al., 2014*).

Single-cell RNA sequencing (scRNA-seq) technology has yielded new insights into cell diversity and differentiation in various tissues. In TEC biology, previous scRNA-seq studies revealed the stochastic nature of TSA expression in mTECs (*Sansom et al., 2014*; *Meredith et al., 2015*) and high heterogeneity of TECs in mice (*Bornstein et al., 2018*; *Miller et al., 2018*; *Dhalla et al., 2020*; *Baran-Gale et al., 2020*). Bornstein et al. showed that mTECs in the postnatal thymus are separated into four subsets, mTEC I to IV (*Bornstein et al., 2018*). In addition to the classical mTEC$^{lo}$ (mTEC I), Aire$^+$ mTEC (mTEC II), and post-Aire mTEC (mTEC III) types, a tuft-like mTEC subset (mTEC IV) was identified (*Bornstein et al., 2018*; *Miller et al., 2018*). Subsequent scRNA-seq studies suggested further heterogeneity of TECs, such as cilium TECs (*Dhalla et al., 2020*), GP2$^+$ TECs (*Dhalla et al., 2020*), intertypical TECs (*Baran-Gale et al., 2020*), neural TECs (*Baran-Gale et al., 2020*), and structural TECs (*Baran-Gale et al., 2020*), according to specific gene expression profiles. However, it has not yet been clarified whether this heterogeneity identified from gene expression profiles is correlated with differences in chromatin structure.

In general, transit-amplifying cells (TACs) are a proliferating cell population linking stem cells and differentiated cells (*Lajtha, 1979*). TACs are short-lived and undergo differentiation after a few cell divisions. To date, the presence of TACs has been confirmed in some tissues such as intestines (*Clevers, 2013*), hair follicles (*Hsu et al., 2014*), and neurons (*Lui et al., 2011*). Previous analyses of scRNA-seq data of murine adult TECs revealed a cell cluster expressing an abundance of cell cycle-regulated genes, which implies the presence of TACs for TECs (TA-TECs) (*Dhalla et al., 2020*; *Wells et al., 2020*). Computational trajectory analysis of scRNA-seq data suggested that this population might give rise to Aire-expressing mTECs (*Dhalla et al., 2020*; *Baran-Gale et al., 2020*). Intriguingly, another trajectory study predicted that this cell cluster might differentiate into Aire-expressing mTECs and an mTEC population expressing CCL21a (*Wells et al., 2020*). However, because the TA-TEC candidate has not been isolated and specific marker genes of TA-TECs have not been reported, exact properties of TA-TECs, in addition to their cellular fates, remain to be clarified.

In this study, droplet-based scRNA-seq and single-cell assays for transposase-accessible chromatin sequencing (scATAC-seq) of murine TECs were performed to characterize TEC heterogeneity and differentiation dynamics. Integrative analysis of these data showed that Aire$^+$ mTECs are separated into at least two clusters with different gene expression profiles and chromatin accessibilities. One of these Aire$^+$ mTEC clusters exhibited high expression of cell cycle-related genes, which accords with

a previously proposed TAC population of mTECs (*Dhalla et al., 2020*; *Wells et al., 2020*). By using the Fucci technology (*Mort et al., 2014*), proliferating mTECs expressing Aire and maturation marker CD80 were isolated as TA-TEC candidates. This proliferating Aire$^+$ CD80$^{hi}$ mTEC subpopulation showed minimal expression of TSAs regulated by AIRE, in contrast to quiescent Aire$^+$ CD80$^{hi}$ mTECs. Moreover, in vivo BrdU pulse-labeling, and in vitro reaggregated thymic organ culture suggested that proliferating Aire$^+$ CD80$^{hi}$ mTECs are short-lived and that they differentiate into quiescent Aire$^+$ CD80$^{hi}$ mTECs, post-Aire mTECs, and tuft-like mTECs. Consequently, these data strongly suggest that proliferating Aire$^+$ CD80$^{hi}$ mTECs are TACs for mTECs expressing TSAs.

## Results
### Droplet-based scATAC-seq reveals heterogeneity of TEC chromatin structure

Given that chromatin structures can be changed during cell differentiation, scATAC-seq analysis of TECs may offer some insights into TEC heterogeneity and differentiation dynamics. Droplet-based scATAC-seq analysis was carried out with EpCAM$^+$ CD45$^-$ cells that were sorted and pooled from thymi of two mice, 4 weeks of age. scATAC-seq analysis was repeated twice and integrated with removal of batch effects via a combination of the Signac R package (*Stuart et al., 2021*) and the Harmony algorithm (*Korsunsky et al., 2019*). Unsupervised graph-based clustering and dimensional reduction via uniform manifold approximation and production (UMAP) revealed 11-cell clusters from 15,255 cells (7884 for Experiment #1 and 7371 for Experiment #2) (*Figure 1A*). Chromatin accessibility of previously known TEC marker genes (gene coordinates including their 2 kbp upstream region) was analyzed. Clusters 0, 4, 5, 6, 8, 9, and 11 contained relatively higher numbers of cells having the open chromatin structure of the *Cd80* gene, a maturation marker of TECs (*Figure 1B and C*). Among these clusters, the *cis*-regulatory element of the *Aire* gene (*LaFlam et al., 2015*) (about 2 kbp upstream from the transcriptional start site) is opened in clusters 0 and 4 (*Figure 1D*), suggesting that these clusters may be concordant with Aire-expressing mTECs (Aire$^+$ mTECs, also referred to as mTEC II *Bornstein et al., 2018*). In contrast, the *cis* element of *Aire* genes is closed in clusters 5, 6, 8, 9, and 11 (*Figure 1D*), suggesting that these clusters may correspond to post-Aire mTECs and other Aire-negative mature mTECs (*Bornstein et al., 2018*). Because the *Irga2* gene (also called Lrmp) region is accessible in cluster 6 (*Figure 1B* and *Figure 1—figure supplement 1*), this cluster may be equivalent to tuft-like mTECs (mTEC IV) (*Bornstein et al., 2018*; *Miller et al., 2018*). CD80 and Aire gene regions in clusters 1, 2, and 3 are relatively closed, whereas the mTEC marker *Tnfrsf11a* is relatively accessible (*Figure 1B and C*, and *Figure 1—figure supplement 1*). Therefore, these clusters should be equivalent to mTECs expressing low levels of CD80 and Aire (mTEC$^{lo}$). Cluster 7 should be cTECs, because a cTEC marker *Psmb11* gene region is opened (*Figure 1B* and *Figure 1—figure supplement 1*). Finally, cluster 10 was deemed thymocyte contamination because the *Rag1* gene was opened (*Figure 1—figure supplement 1*). Comparison between two independent scATAC-seq experiments suggested relatively high reproducibility (*Figure 1—figure supplement 2*).

We next sought to correlate scATAC data with TEC scRNA-seq data. Droplet-based scRNA-seq analysis of EpCAM$^+$CD45$^-$ cells from age- and gender-matched mice (4-week female mice) was performed. scRNA-seq analysis was repeated twice and integrated with the removal of batch effects via the Seurat package (*Butler et al., 2018*). Analysis of integrated data (total 11,475 cells) revealed 18-cell clusters in the UMAP dimension (*Figure 2A*, *Figure 2—figure supplement 1*, and *Supplementary file 1*). Comparison between two independent experiments suggested high reproducibility of the scRNA-seq analysis (*Figure 2—figure supplement 2*).

These TEC clusters were assigned according to expression of TEC marker genes (*Figure 2* and *Figure 2—figure supplement 3*). Clusters R0, R1, R3, and R9 showed high expression of *Aire* (*Figure 2B*), suggesting that these clusters are equivalent to Aire$^+$ mTECs (also referred to as mTECs II). Clusters R2, R4, and R5 include cells showing relatively higher levels of *Itga6* and *Ccl21a* expression and a very low level of *Aire* expression (*Figure 2B*), corresponding to mTEC I (*Bornstein et al., 2018*), CCL21-expressing mTECs (*Lucas et al., 2020*), and possibly intertypical TECs (*Baran-Gale et al., 2020*). Cluster R6 expresses *Irga2* (*Figure 2B*) and should contain tuft-like mTECs (mTEC IV) (*Bornstein et al., 2018*). Clusters R7 and R10 were marked with *Krt10* and *Pigr* genes, respectively (*Figure 2B*). Accordingly, these clusters should be categorized as post-Aire mTECs (also referred

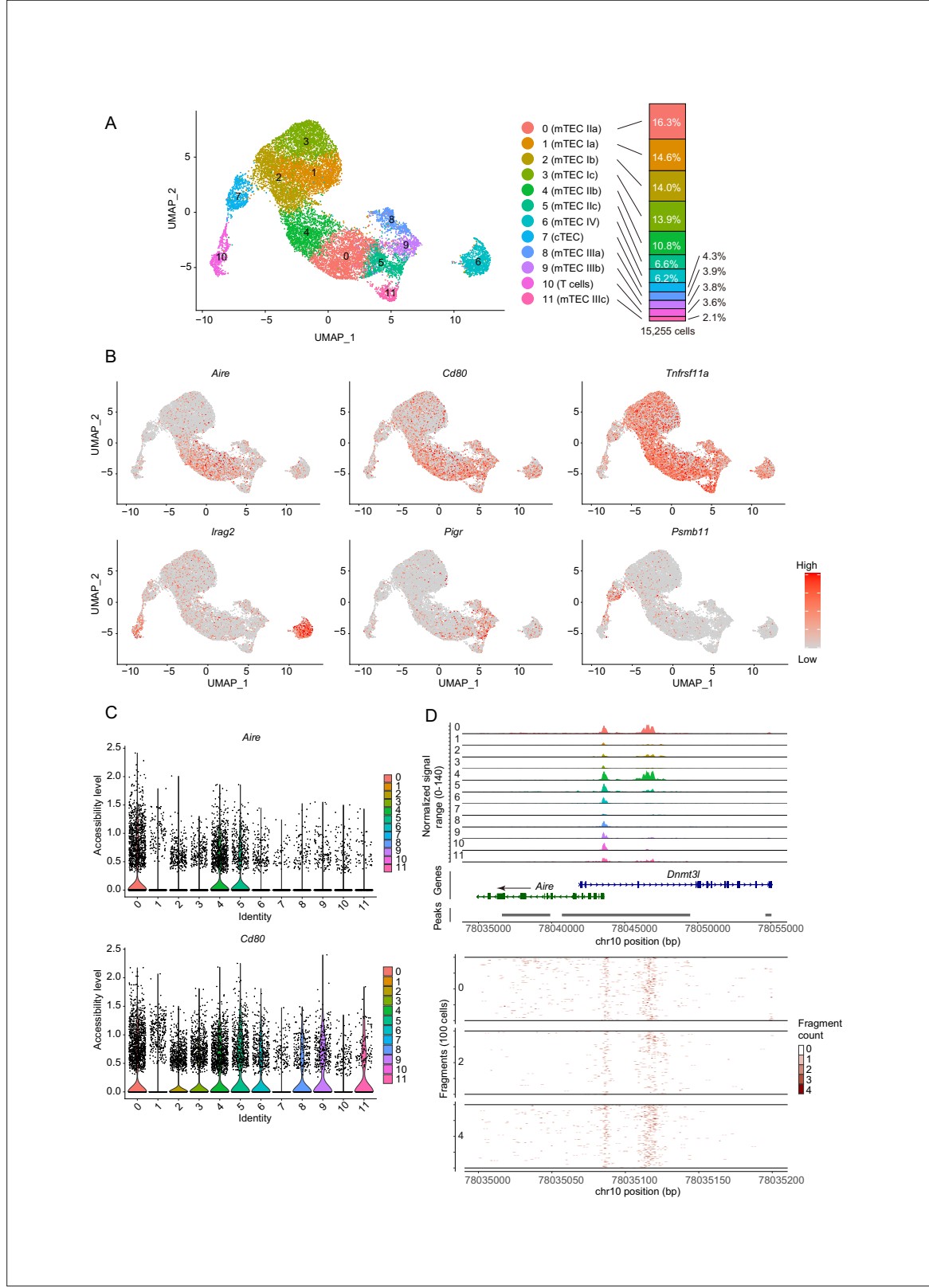

**Figure 1.** Droplet-based single-cell assays for transposase-accessible chromatin sequencing (scATAC-seq) analysis of thymic epithelial cells (TECs) in 4-week mice. (**A**) Uniform manifold approximation and production (UMAP) plot of scATAC-seq data from TECs (EpCAM⁺ CD45⁻ TER119⁻) from 4-week mice. Cell clusters are separated by colors and numbers in the plot. The two datasets were integrated using the Seurat package. The graph on the right shows percentages of each cluster in the total number of cells detected (15,255 cells). (**B**) Chromatin accessibility of typical marker genes of TECs.

*Figure 1 continued on next page*

*Figure 1 continued*

Accessibility in each gene region is represented in red. (**C**) Violin plot depicting chromatin accessibility in *Aire* and *Cd80* gene regions in each cluster. (**D**) Pseudo-bulk accessibility tracks of the *Aire* gene region in each cluster (upper panels) and frequency of sequenced fragments within the *Aire* gene region of individual cells in cluster 0, 2, and 4 (lower panels).

The online version of this article includes the following figure supplement(s) for figure 1:

**Figure supplement 1.** Violin plot of chromatin accessibility in thymic epithelial cell (TEC) marker gene regions in each cluster.

**Figure supplement 2.** Comparison of uniform manifold approximation and production (UMAP) plots of single-cell assays for transposase-accessible chromatin sequencing (scATAC-seq) between two experiments.

**Figure supplement 3.** Comparison of single-cell assays for transposase-accessible chromatin sequencing (scATAC-seq) data between two experiments.

to as mTECs III *Bornstein et al., 2018*). Cluster R13 showed high expression of chemokines, *Ccl6* and *Gp2* (*Figure 2B* and *Figure 2—figure supplement 3*), which is concordant with Gp2⁺ TECs, as reported recently (*Dhalla et al., 2020*). Clusters R8 and R11 exhibited high expression of typical cTEC marker genes, *Psmb11* and *Dll4* (*Figure 2B* and *Figure 2—figure supplement 3*), and should be equivalent to cTECs. Given that thymocyte genes are highly expressed, cluster R11 was most likely thymic nurse cells enclosing thymocytes (*Nakagawa et al., 2012*). Cluster R12 showed relatively high expression of *Pdpn* (*Figure 2—figure supplement 3*), which may comprise junctional TECs (*Onder et al., 2015*). Cluster R14 was considered thymocyte contamination because thymocyte markers, but not TEC markers, were detected. Cluster R15 apparently corresponds to structural TECs, reported recently, because of their expression of *Car8* and *Cd177* (*Baran-Gale et al., 2020*; *Figure 2—figure supplement 3*). Cells in cluster R16 highly express *Tppp3* and *Fam183b* (*Figure 2—figure supplement 3*). Since these genes are expressed in ciliated cells (*Orosz and Ovádi, 2008*; *Beckers et al., 2018*), this cluster may be equivalent to ciliated columnar TECs associated with thymic cystic structure (*Dhalla et al., 2020*; *Khosla and Ovalle, 1986*; *Park et al., 2020*). We failed to assign cluster R17, which may be contaminated with endothelial cells or macrophages, because they express *Ly6c1* and *Aqp1*, but low levels of *Epcam* (*Figure 2—figure supplement 3*). Correlation of our scRNA-seq data with reported scRNA-seq data (*Bornstein et al., 2018*; *Dhalla et al., 2020*; *Baran-Gale et al., 2020*; *Wells et al., 2020*) was investigated by integrating datasets (*Figure 2—figure supplement 4*) or checking expression of differentially expressed genes in clusters of other datasets (*Figure 2—figure supplements 5 and 6*). Overall, our data and assignments were reasonably correlated with previous scRNA-seq data analyses.

In next, we bioinformatically integrated the scRNA-seq data with scATAC-seq data. Gene expression, predicted from accessible chromatin regions of scATAC-seq data, was correlated with scRNA-seq data using the Signac R package (*Figure 3* and *Figure 3—figure supplement 1*). As described, clusters 0 and 4 in scATAC-seq analysis contain cells with the accessible *cis*-regulatory element of the *Aire* gene (*Figure 1D*). Consistently, cluster 0 in scATAC-seq was mostly transferred to clusters R0 (56.8 %) and R3 (12.9%) in scRNA-seq analysis (*Figure 3B and C*, and *Supplementary file 2*), which were assigned as Aire⁺ mTECs (*Figure 2*). Cells transferred to R0 and R3 appear to be separately embedded in cluster 0 in the UMAP dimension, implying that these Aire⁺ mTEC subsets have slightly different chromatin structures. Cluster 4 was mostly transferred to cluster R1 (57.1%) (*Figure 3B and C*), also designated as Aire⁺ mTECs, and in part R2 (23.5%), which belongs to mTEC I. Interestingly, cells transferred to cluster R9 are embedded around the junction between cluster 0 and 4 (*Figure 3B*), suggesting that cluster R9 may be a transitional stage between R1 and R0. Clusters 1, 2, and 3 are closely embedded in the UMAP dimension and principally assigned to clusters R2, R4, and R5 (*Figure 3B and C*), suggesting that these clusters are concordant with mTEC I or intertypical TECs assigned in the scRNA-seq data. Cluster 5 mainly contains cells transferred to cluster R3 (56.0%) and R10 (27.4%) (*Figure 3B*), which were assigned as late-Aire mTECs (mTEC IIc) and post-Aire mTECs (mTEC III), respectively. As expected, cluster 6 with an open *Irga2* gene was transferred to cluster R6, a tuft-like mTEC subset (mTEC IV). Cluster 7 was transferred to cluster R8 and R12, assigned as cTECs and jTECs, respectively. Cluster 9 was assigned as cluster R7, which is Krt10⁺ mTEC III subset (*Figure 3C*). Cluster 8 contains clusters R15 (35.4%) and R16 (12.2%), which express markers of structural TECs and cilia TECs, respectively (*Figure 3C* and *Figure 3—figure supplement 1*), in addition to R7 and R10 assigned as mTEC III. Cluster 11 was concordant with R13, which was Gp2⁺ TECs (*Figure 3B and C*). Finally, cluster 10 was transferred to clusters R11 and R14, which are assigned as

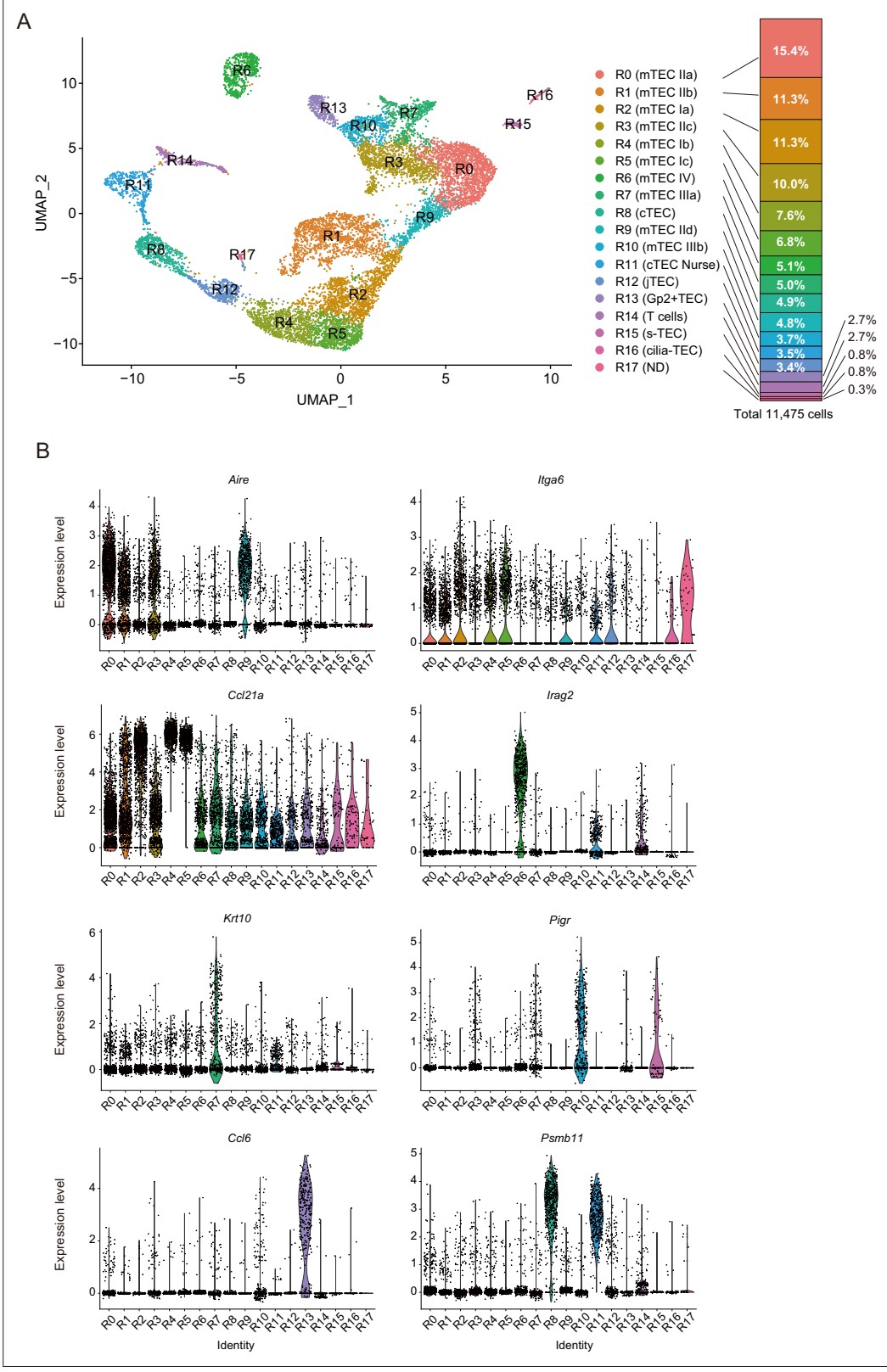

**Figure 2.** Droplet-based single-cell RNA sequencing (scRNA-seq) analysis of thymic epithelial cells (TECs) in 4-week mice. (**A**) Uniform manifold approximation and production (UMAP) plot of scRNA-seq data from TECs (EpCAM⁺ CD45⁻ TER119⁻) from 4-week mice. Cell clusters (R0 to R17) are indicated by colors and numbers in

*Figure 2 continued on next page*

*Figure 2 continued*

the plot. The graph on the right shows the percentages of each cluster in the total number of cells detected (11,792 cells). (**B**) Violin plots depicting expression levels of typical TEC marker genes in each cluster.

The online version of this article includes the following figure supplement(s) for figure 2:

**Figure supplement 1.** Heatmap of the top five genes selectively expressed in each subcluster.

**Figure supplement 2.** Comparison of single-cell RNA sequencing data between two experiments.

**Figure supplement 3.** Violin plots of marker genes and cell cycle-related genes.

**Figure supplement 4.** Comparison of single-cell RNA sequencing data between this study and previous studies.

**Figure supplement 5.** Dot plots of expression of marker genes for our single-cell RNA sequencing (scRNA-seq) clusters in clusters of other scRNA-seq datasets (*Bornstein et al., 2018*; *Dhalla et al., 2020*; *Wells et al., 2020*).

**Figure supplement 6.** Dot plots of expression of marker genes for Baran-Gales's single-cell RNA sequencing (scRNA-seq) clusters (*Baran-Gale et al., 2020*) in clusters of our scRNA-seq datasets.

T cells and nurse TECs (*Figure 3C* and *Figure 3—figure supplement 1*). Although a few cells were transferred to R17 in scRNA-seq, these cells did not form cluster in this analysis. These assignments were practically the same in the two scATAC datasets (*Figure 3—figure supplement 2*). Overall, the integration analysis suggested that TEC heterogeneity predicted from scRNA-seq may be ascribed to differences in chromatin structure.

## Aire-positive mTECs are divided into two subsets having distinct chromatin structures

Previous scRNA-seq studies proposed the existence of a TEC population showing high expression of cell cycle-regulated genes (*Dhalla et al., 2020*; *Baran-Gale et al., 2020*; *Wells et al., 2020*). In our scRNA-seq data, cluster R1 (mTEC IIb) appears equivalent to such a TEC subset, expressing cell cycle-related genes (*Figure 2—figure supplement 3B*). Integrative analysis of scRNA-seq and scATAC-seq suggested that cells in cluster 4 in scATAC-seq were mainly transferred to cluster R1 (*Figure 3B*). Although both clusters 4 and 0 have the accessible enhancer element of the Aire gene (*Figure 1*), 269 genomic regions were significantly opened, and 147 regions were closed in cluster 4, in contrast to cluster 0 (*Supplementary file 3* and *Figure 3—figure supplement 3*). Thus, it is likely that the Aire[+] mTECs are divided into two subsets based on expression of proliferation markers and chromatin accessibility.

RNA velocity, which recapitulates differentiation dynamics by comparing unspliced and spliced RNA in scRNA-seq data (*La Manno et al., 2018*), predicted that cluster R1 may differentiate into other Aire[+] mTECs (clusters R0, R3, and R9) (*Figure 3—figure supplement 4A*), which is consistent previous analyses (*Dhalla et al., 2020*). Moreover, trajectory analysis of scATAC-seq data using Monocle3 suggested a possible transition between cluster 4 and cluster 0 (*Figure 3—figure supplement 4B*). Overall, integrative analysis of scATAC-seq and scRNA-seq data imply that the Aire[+] mTEC subset expressing cell cycle-related genes (cluster 4 in scATAC-seq and cluster R1 in scRNA-seq) may be equivalent to transiently amplifying cells (TA cells) with a distinct chromatin structure. Although a previous study suggested a trajectory of proliferating TEC clusters to mTEC I[31], RNA velocity of our scRNA-seq data did not clearly recapitulate it (*Figure 3—figure supplement 4A*).

## A proliferating TEC cluster is sub-divided into an Aire-expressing subcluster and the Aire-negative Ccl21a[high] subcluster

Subclustering of cluster R1 showed its separation into seven subclusters (R1A to R1G in *Figure 4A and B*). Clusters R1A, R1B, R1C, R1D, and R1E showed expression of *Aire* and *Cd80* (*Figure 4C and D*). In contrast, *Ccl21a*, but not *Aire*, is highly expressed in clusters R1F and R1G (*Figure 4C and D*), which may be consistent with a previous study (*Wells et al., 2020*). Interestingly, cell cycle scoring analysis suggested that R1A, R1B, and R1F contain mainly G2M phase cells. In contrast, R1D, R1E, and R1G contain S phase cells and R1C contains both G2M and S phase cells (*Figure 4E and F*). Thus, the *Aire*[+] subclusters and the *Aire*-negative *Ccl21a*[high] subclusters may be further divided by cell cycle phase. These data suggested that proliferating TECs consist of two distinct subsets, distinguished by expression levels of *Aire* and *Ccl21a*. Correlation between subclusters of R1 (R1A to R1G) and scATAC-seq

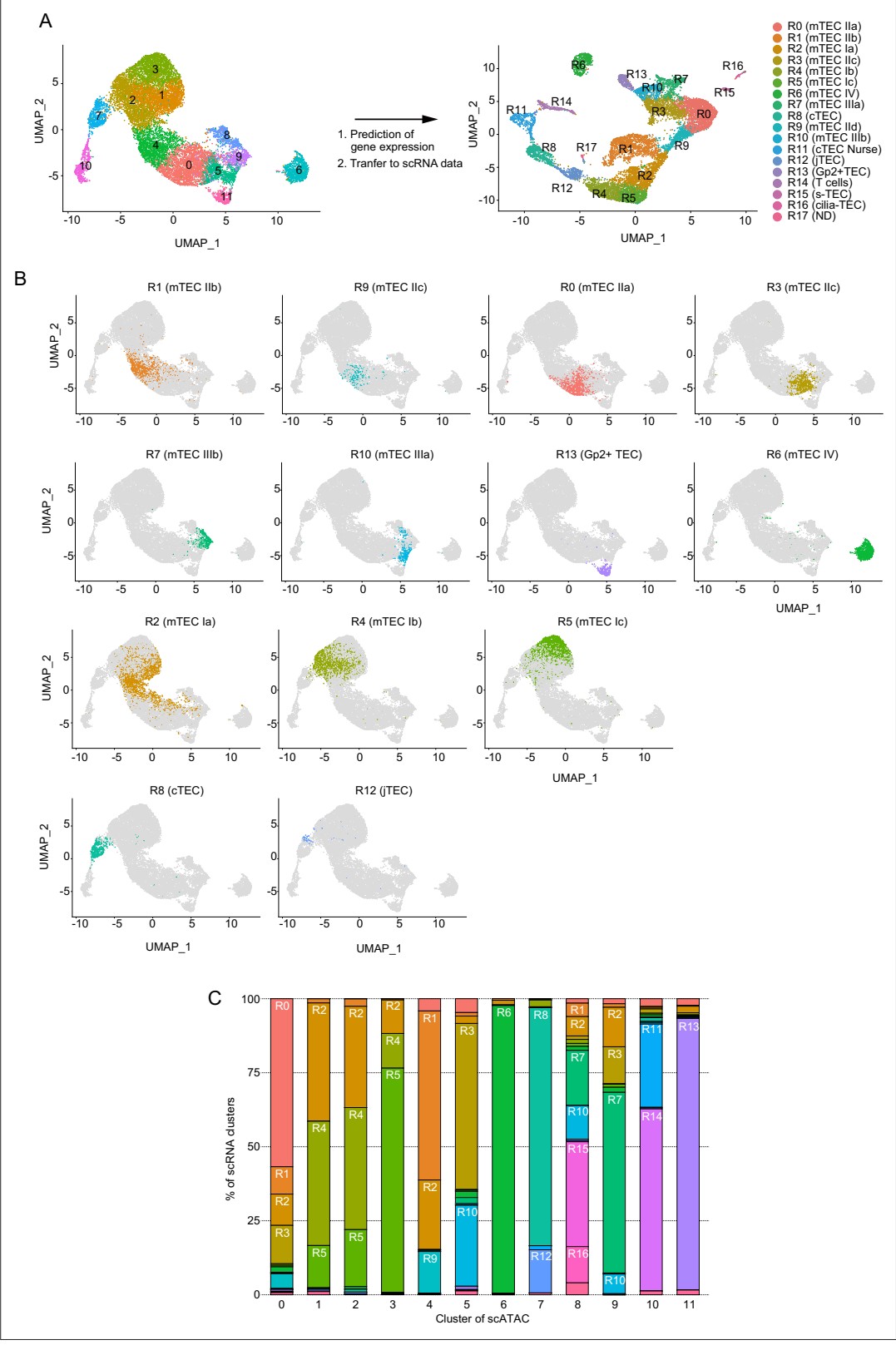

**Figure 3.** Integrative analysis of single-cell assays for transposase-accessible chromatin sequencing (scATAC-seq) data and single-cell RNA sequencing (scRNA-seq) data of thymic epithelial cells (TECs). (**A**) Gene expression was predicted from scATAC-seq data using Signac. Individual cells in the cluster from scATAC data (clusters 0 to 11) were assigned and transferred to the uniform manifold approximation and production (UMAP) plot of scRNA-

*Figure 3 continued on next page*

*Figure 3 continued*

seq cluster (R0 to R17). (**B**) Correlation between clusters derived from scATAC-seq and scRNA-seq datasets of TECs. Cell types were annotated in scATAC dataset of TECs by transferring clusters from an scRNA-seq dataset. (**C**) Ratios of cells assigned to each scRNA-seq cluster in each scATAC cluster.

The online version of this article includes the following figure supplement(s) for figure 3:

**Figure supplement 1.** Integrative analysis of single-cell assays for transposase-accessible chromatin sequencing (scATAC-seq) data and single-cell RNA sequencing (scRNA-seq) data of thymic epithelial cells (TECs).

**Figure supplement 2.** Comparison of Integrative analysis of single-cell assays for transposase-accessible chromatin sequencing (scATAC-seq) data and single-cell RNA sequencing (scRNA-seq) data of thymic epithelial cells (TECs) between two experiments.

**Figure supplement 3.** Pseudo-bulk accessibility tracks and frequency of sequenced fragments.

**Figure supplement 4.** Trajectory analysis of single-cell RNA sequencing data.

clusters was analyzed (*Figure 4—figure supplement 1*). Cells in cluster 4 were transferred to both the *Aire*$^+$ (R1A to R1E) and the *Aire*-negative *Ccl21a*$^{high}$ (R1F and R1G) subclusters (*Supplementary file 4*), implying that subclusters in R1 may have similar chromatin accessibility, although this point should be explored further in the future.

## A proliferative cell subset is present in Aire$^+$ mTECs

TA cells were generally defined as a proliferative, short-lived cell subset generated from progenitor or stem cells and differentiating into mature quiescent cells (*Lajtha, 1979*; *Zhang and Hsu, 2017*). Analysis of scRNA-seq and scATAC-seq data suggested that proliferating Aire$^+$ mTECs may be equivalent to TA cells with distinct chromatin accessibility. To search for evidence supporting the presence of TA cells of mTECs (TA-TECs), we first sought to isolate the proliferating Aire$^+$ CD80$^{hi}$ mTEC subset as candidate TA-TECs. Fucci2a mice, in which cell cycle progression can be monitored with mCherry (G1 and G0 phases) and Venus (G2, M, and S phases) fluorescence, were used to isolate such proliferating cells (*Figure 5A*; *Mort et al., 2014*; *Lazzeri et al., 2018*; *Wong et al., 2018*; *Antonica et al., 2019*), and were crossed with Aire-GFP-reporter mice to facilitate detection of Aire expression (*Yano et al., 2008*). Flow cytometric analysis indicated that Venus$^+$ cells are present among mTECs expressing high levels of CD80 (mTEC$^{hi}$) (*Figure 5—figure supplement 1A*), although the expression level of CD80 might be slightly lower than that of CD80$^+$ mCherry$^{hi}$ mTECs (*Figure 5—figure supplement 1B*). Moreover, these Venus$^+$ mTEC$^{hi}$ cells expressed GFP (*Figure 5B*), indicating expression of AIRE. Thus, these data revealed the presence of dividing cells in the Aire$^+$ CD80$^{hi}$ mTEC fraction. The fluorescence intensity of Aire-GFP in Venus$^+$ CD80$^{hi}$ mTECs showed a broad peak and was slightly lower than that of Venus$^-$ mTEC$^{hi}$ cells, which may be due to the relatively lower expression of Aire in Venus$^+$ CD80$^{hi}$ mTECs. However, the compensation between GFP and Venus proteins, which have very close fluorescence spectra, hampered an exact comparison of Aire expression levels between Venus$^+$ mTEC$^{hi}$ cells and Venus$^-$ mTEC$^{hi}$ cells. We next confirmed Aire protein expression in proliferating mature mTECs. Immunostaining with an anti-Aire-antibody revealed the presence of Aire protein localized in the nucleus of sorted Venus$^+$ CD80$^{hi}$ mTECs (*Figure 5C*). Immunostaining of the thymic section from *Foxn1*-specific Fucci2a mice revealed that Venus$^+$ cells are localized in the medulla, and some of the Aire$^+$ mTECs were Venus$^+$ (*Figure 5D* and *Figure 5—figure supplement 2*). Taken together, these data confirm the presence of proliferating Aire$^+$ CD80$^{hi}$ mTECs in the thymic medulla.

## Proliferating Aire$^+$ mTECs express low levels of Aire-dependent TSAs

We next addressed whether the proliferating Aire$^+$CD80$^{hi}$ mTECs subset has a molecular signature distinct from that of quiescent Aire$^+$CD80$^{hi}$ mTECs. RNA-seq analysis of sorted cells from Fucci mice suggested that Venus$^+$ CD80$^{hi}$ mTECs and Venus$^-$ CD80$^{hi}$ mTECs subsets have considerably different gene expression profiles (*Figure 5E*). As expected, gene ontology analysis confirmed enrichment of cell cycle-related genes in Venus$^+$ CD80$^{hi}$ mTECs compared with Venus$^-$ CD80$^{hi}$ mTECs (*Supplementary file 5*). Notably, the Venus$^+$ CD80$^{hi}$ mTEC subset expressed lower levels of Aire-dependent TSAs (*Supplementary file 6*) than the Venus$^-$ CD80$^{hi}$ mTECs subset (*Figure 5F and G* and *Figure 5—figure supplement 3*). Whereas expression of Aire-independent TSAs was also low in the Venus$^+$ CD80$^{hi}$ mTEC subset, the difference was smaller than in Aire-dependent TSAs (*Figure 5G* and *Figure 5—figure*

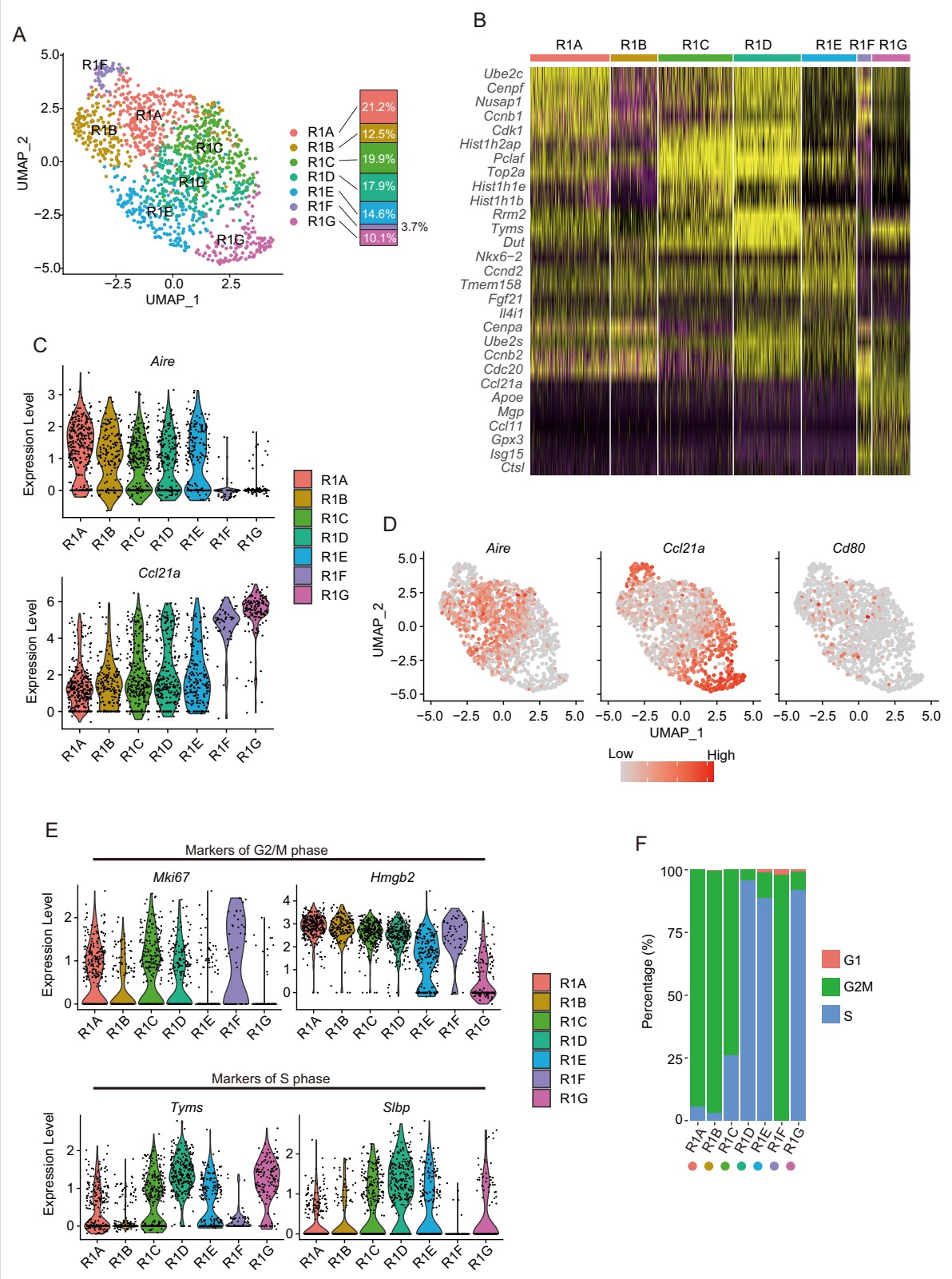

**Figure 4.** Subcluster analysis of the thymic epithelial cell (TEC) subset expressing a high level of cell cycle-related genes. (**A**) Uniform manifold approximation and production (UMAP) plot of single-cell RNA sequencing (scRNA-seq) data of each subcluster (R1A to R1G) in R1. Cell subclusters (R1A to R1G) are separated by colors and numbers in the plot. The graph on the right shows the percentages of each cluster in the parent R1 cluster. (**B**) Heatmap of the top five genes selectively expressed in each subcluster. Yellow color indicates high expression. (**C**) Expression levels of *Aire*

*Figure 4 continued on next page*

*Figure 4 continued*

and *Ccl21a* in the subcluster are exhibited as violin plots. (**D**) Expression levels of *Aire*, *Ccl21a*, and *Cd80* in the subcluster are shown in dot plots.
(**E**) Expression levels of marker genes for G2/M phase (upper, *Mki67* and *Hmgb2*) and S phase (lower, *Tyms* and *Slbp*) in the subcluster are exhibited as violin plots. (**F**) Percentage of cells predicted as each cell cycle (G1, G2M, and S phases) in the subclusters.

The online version of this article includes the following figure supplement(s) for figure 4:

**Figure supplement 1.** Integrative analysis of cluster 4 from single-cell assays for transposase-accessible chromatin sequencing (scATAC-seq) data and subclusters in R1 from single-cell RNA sequencing (scRNA-seq) data.

*supplement 3*). These data suggested that proliferating Aire⁺CD80^hi mTECs are phenotypically immature, compared to quiescent Aire⁺CD80^hi mTECs.

## Proliferating Aire⁺ mTECs are precursors of mature mTECs

Because TA cells are defined as short-lived cells differentiating into mature cells (*Lajtha, 1979*), we next addressed this issue regarding the proliferating Aire⁺ CD80^hi mTECs. In vivo pulse labeling of TECs with 5-bromo-2′-deoxyuridine (BrdU) was performed. Because mCherry^hi cells and mCherry^lo were generally in G0 and G1 stages of the cell cycle, respectively (*Tomura et al., 2013*), each fraction in CD80^hi mTECs was sorted separately after *i.p.* administration of BrdU, and thereafter stained with anti-BrdU antibody (*Figure 6A*). This procedure was necessary because mCherry fluorescence is lost after BrdU staining. Flow cytometric analysis showed that approximately 35% of mCherry^lo CD80^hi mTECs (hereafter referred to as mCherry^lo) were labeled 12 hr (day 0.5) after the BrdU injection (*Figure 6B*). In contrast, about 3% of mCherry^hi CD80^hi mTECs (referred to as mCherry^hi) were BrdU-positive (*Figure 6B*). Thus, as expected, cell cycle progression of mCherry^lo is much faster than mCherry^hi. Importantly, cell number and the ratio of BrdU-positive cells in the mCherry^lo fraction were significantly decreased 3.5 days after the BrdU injection (*Figure 6B and C*). On the other hand, the frequency of BrdU-positive cells in mCherry^hi was increased by day 3.5, and plateaued from day 3.5 to day 6.5 (*Figure 6B and C*). Notably, mean fluorescence intensity (MFI) of BrdU staining in mCherry^hi at day 3.5 was about 50% lower than that in mCherry^lo at day 0.5 (*Figure 6D*), suggesting that mCherry^hi cells at day 3.5 were generated after cell division. Overall, these data suggest that mCherry^lo cells are transiently proliferating, and after cell division, they are converted to mCherry^hi, having low proliferative activity.

To verify that mCherry^lo cells are precursors of mCherry^hi, we performed an in vitro reaggregated thymic organ culture (RTOC) experiment (*Figure 7A*). The mCherry^lo fraction was sorted (*Figure 7—figure supplement 1A*) and reaggregated with wild type embryonic thymic cells. After 5 days of culture, mCherry^hi was detected in RTOC (*Figure 7A*). Because Venus⁺mCherry^lo cells were practically absent in RTOC (*Figure 7—figure supplement 1B*), surviving mCherry^lo cells were mostly converted into mCherry^hi in RTOC. The possibility that mCherry^hi contaminating cells during cell sorting survived in RTOC was ruled out by control reaggregation experiments using only mCherry^hi (*Figure 7—figure supplement 1C*) in addition to the higher expression of pro-apoptotic genes (*Liberzon et al., 2015*) in mCherry^hi than mCherry^lo (*Figure 7—figure supplement 1D*). Interestingly, reaggregation with allogenic fetal thymus (Balb/cA background) was not sufficient for conversion to mCherry^hi (*Figure 7—figure supplement 1E*), implying that high affinity interaction between TCR and MHC contributes to survival and maintenance of mCherry^lo TECs, as described previously (*Irla et al., 2008*). Next, we sorted mCherry^hi cells in the RTOC (referred to as mCherry^hi-RTOC) in addition to mCherry^lo and mCherry^hi from the Fucci thymus, and analyzed gene expression by RNA-seq. As expected, the mCherry^lo fraction expressed a lower level of Aire-dependent TSAs, compared to mCherry^hi (*Figure 7B* and *Figure 7—figure supplement 2A*), although Aire and Mki67 were highly expressed (*Figure 7C* and *Figure 7—figure supplement 2A*). Importantly, in comparison to the mCherry^lo fraction, the mCherry^hi-RTOC fraction showed higher levels of Aire-dependent TSAs (*Figure 7B*). Moreover, beside cell cycle-related genes, some genes were highly expressed in all mCherry^lo, Venus⁺ cells, and cluster R1 cells (*Figure 7—figure supplement 2B* and *Supplementary file 7*). Notably, these gene set were down-regulated in mCherry^hi-RTOC (*Figure 7D* and *Figure 7—figure supplement 2C*). These data suggest that mCherry^lo cells were converted into mCherry^hi in RTOC.

In order to detail phenotypes of mCherry^hi-RTOC, we next performed well-based scRNA-seq. mCherry^hi-RTOC, in addition to mCherry^loCD80^hi and mCherry^hiCD80^hi mTECs from the murine thymus, were single-cell sorted by flow cytometry, and then gene expression in individual cells was

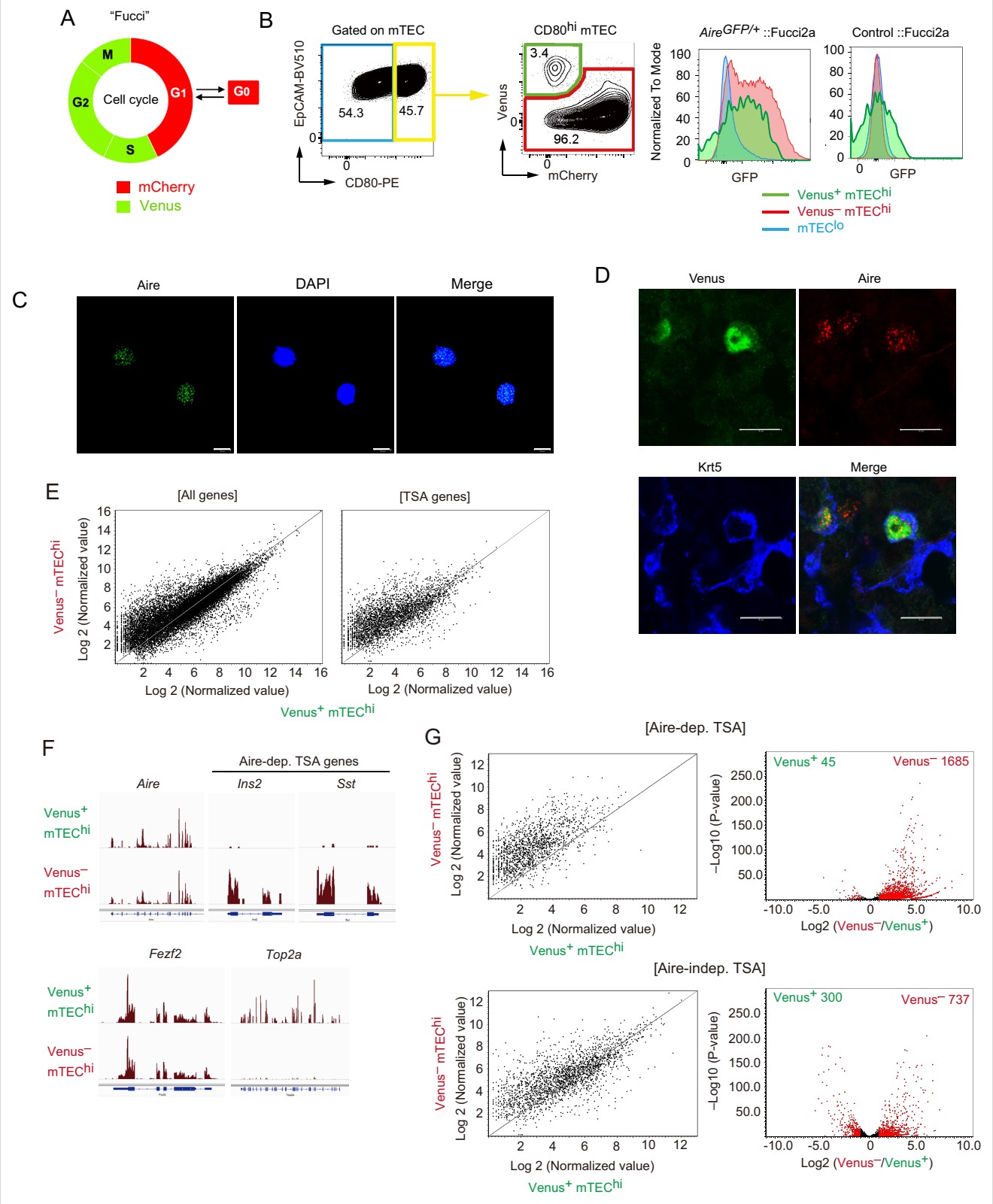

**Figure 5.** A highly proliferative subset of Aire⁺ CD80$^{hi}$ medullary thymic epithelial cells (mTECs). (**A**) Schematic depiction of cell cycles and Fucci fluorescence. (**B**) Flow cytometric analysis of TECs from Fucci2a mice crossed with Aire-GFP-reporter mice. The gating strategy is shown. Intensities of GFP to monitor Aire expression in each subset (Venus⁺ CD80$^{hi}$ mTEC, Venus⁻ CD80$^{hi}$ mTEC, and CD80$^{lo}$ mTEC$^{lo}$) are shown in the right panels. Left, Aire$^{gfp/+}$:: Fucci2a; right, control::Fucci2a. Typical figures of three independent experiments are exhibited. (**C**) Immunostaining of a sorted Venus⁺

Figure 5 continued

CD80<sup>hi</sup> mTEC subset via anti-Aire antibody and DAPI (nucleus staining). Typical panels of three independent experiments are exhibited. Scale bars, 10 μm. (D) Immunostaining of thymic sections from Fucci2a mice with anti-Aire and anti-keratin-5 (Krt5) antibodies. Typical panels of three independent experiments are exhibited. Scale bars, 10 μm. (E) Scatter plots of RNA sequencing data from Venus⁺ CD80<sup>hi</sup> mTEC and Venus⁻ CD80<sup>hi</sup> mTEC subsets. The left panel shows a plot of all detected genes and the right panel shows tissue-specific antigen (TSA) genes detected. N = 3. (F) Atypical RNA sequencing tracks of *Aire*, typical Aire-dependent TSA genes (*Ins* and *Sst*), *Fezf2*, and *Top2a* (a marker of G2/M phase). (G) Scatter plots and volcano plots of RNA sequencing data from Venus⁺ CD80<sup>hi</sup> mTEC and Venus⁻ CD80<sup>hi</sup> mTEC subsets. Upper panels show Aire-dependent TSAs, lower panels show Aire-independent TSAs. Red dots in volcano plots indicate genes for which expression differed significantly (twofold change and FDR p < 0.05) in Venus⁺ and Venus⁻ CD80<sup>hi</sup> mTEC subsets. Numbers of differentially expressed genes are shown in the panels. N = 3. Y axis is log10 of FDR p-value.

The online version of this article includes the following figure supplement(s) for figure 5:

**Figure supplement 1.** Flow cytometric analysis of medullary thymic epithelial cells (mTECs) from Fucci mouse.

**Figure supplement 2.** Immunostaining of thymic sections from Fucci2a mice with anti-GFP (for Venus staining, green) and anti-keratin-5 (Krt5, blue) antibodies.

**Figure supplement 3.** Gene set enrichment analysis of differentially expressed genes between Venus⁻ and Venus⁺ cells.

determined by random displacement amplification sequencing (RamDA-seq) technology (*Hayashi et al., 2018*). After quality control of the data, gene expression matrix data of single-cell RamDA-seq (scRamDA-seq) were integrated with the droplet-based scRNA-seq data (*Figure 7E*). Although this integration slightly changed the UMAP dimension and clustering compared to *Figure 2*, assignment of each cluster was successfully achieved in practically the same fashion (*Figure 7—figure supplement 3*), except that cluster R15 (s-TEC) in *Figure 2* was incorporated into cluster R10 (mTEC IIIb) and one new cluster was separated from cluster R2 and R3.

Cells from the mCherry<sup>lo</sup>CD80<sup>hi</sup> mTEC fraction (total 36 cells) were assigned mainly to clusters R1 (17 cells) and R9 (11 cells) (*Figure 7E and F*, and *Supplementary file 8*). Some cells were assigned to clusters R0 (3 cells) and R2 (2 cells). Although other cells were assigned to clusters R4, R7, and R14, the embedded position was separated from each parent cluster, which may be due to misclustering. In contrast, cells in the mCherry<sup>hi</sup>CD80<sup>hi</sup> mTEC fraction (total 35 cells) were more heterogenous and consisted of cells assigned mainly to clusters R0 (7 cells), R3 (9 cells), R5 (4 cells), R7 (3 cells), R10/15 (5 cells), and R13 (2 cells) (*Figure 7E and F*, and *Supplementary file 8*). Except for cluster R5, these clusters were concordant with Aire⁺ mTECs, post-Aire mTECs, and GP2⁺ TECs. Notably, after RTOC, heterogenous cell populations including clusters R0 (18 cells), R3 (13 cells), R5 (5 cells), R6 (3 cells), R7 (8 cells), and R10/15 (5 cells) were found in the mCherry<sup>hi</sup>-RTOC population (total 65 cells). Its composition was relatively similar to that of the mCherry<sup>hi</sup>CD80<sup>hi</sup> mTEC fraction (*Figure 7F*). Moreover, these mCherry<sup>hi</sup>-RTOC cells expressed high levels of TSAs (*Figure 7G*). Interestingly, 5 cells in mCherry<sup>hi</sup>-RTOC were assigned to cluster R5, which also reside in the mCherry<sup>hi</sup>CD80<sup>hi</sup> mTEC fraction from the adult thymus. This finding is consistent with the idea of an 'intertypical' mTEC cluster, which reportedly contains both CD80<sup>hi</sup> mTECs and CD80<sup>lo</sup>mTECs (*Baran-Gale et al., 2020*). Overall, these data suggest that mCherry<sup>lo</sup>CD80<sup>hi</sup> mTECs differentiate into quiescent mature mTECs expressing high levels of TSAs, including Aire⁺ mTECs (mTEC II), post-Aire mTECs (mTEC III), and tuft-like mTECs (mTEC IV).

## Proliferating Aire⁺ mTECs are present after puberty in mice

We investigated whether proliferative Aire⁺ mTECs persisted in thymi of older mice. TECs were analyzed in 4-, 8-, and 19-week Fucci Aire-GFP mice. Flow cytometric analysis showed that a Venus⁺ mTEC<sup>hi</sup> subset was present in 19-week mice as well as younger mice (*Figure 8A*). Moreover, Venus⁺ mTEC<sup>hi</sup> cells expressed Aire genes (*Figure 8A*). These data strongly suggested that TA-TECs persist in the adult thymus as a source of mature mTECs.

Integrative computational analysis of our scRNA-seq data with a previously reported dataset of fetal TECs (E12 to E18) (*Kernfeld et al., 2018*) showed considerably different cell embedding between adult TECs and fetal TECs (*Figure 8—figure supplement 1*). This trend was common when other scRNA-seq data of adult TECs (*Bornstein et al., 2018*; *Dhalla et al., 2020*; *Wells et al., 2020*) was integrated with the fetal TEC data (*Figure 8—figure supplement 2*). A TEC-expressing subset was present in the fetal thymus whereas Aire expression was low (clusters F3 and F12, *Figure 8—figure supplement 1B*). This implies that fetal proliferating mTECs may have a different gene expression profile than adult proliferating Aire⁺CD80<sup>hi</sup> mTECs.

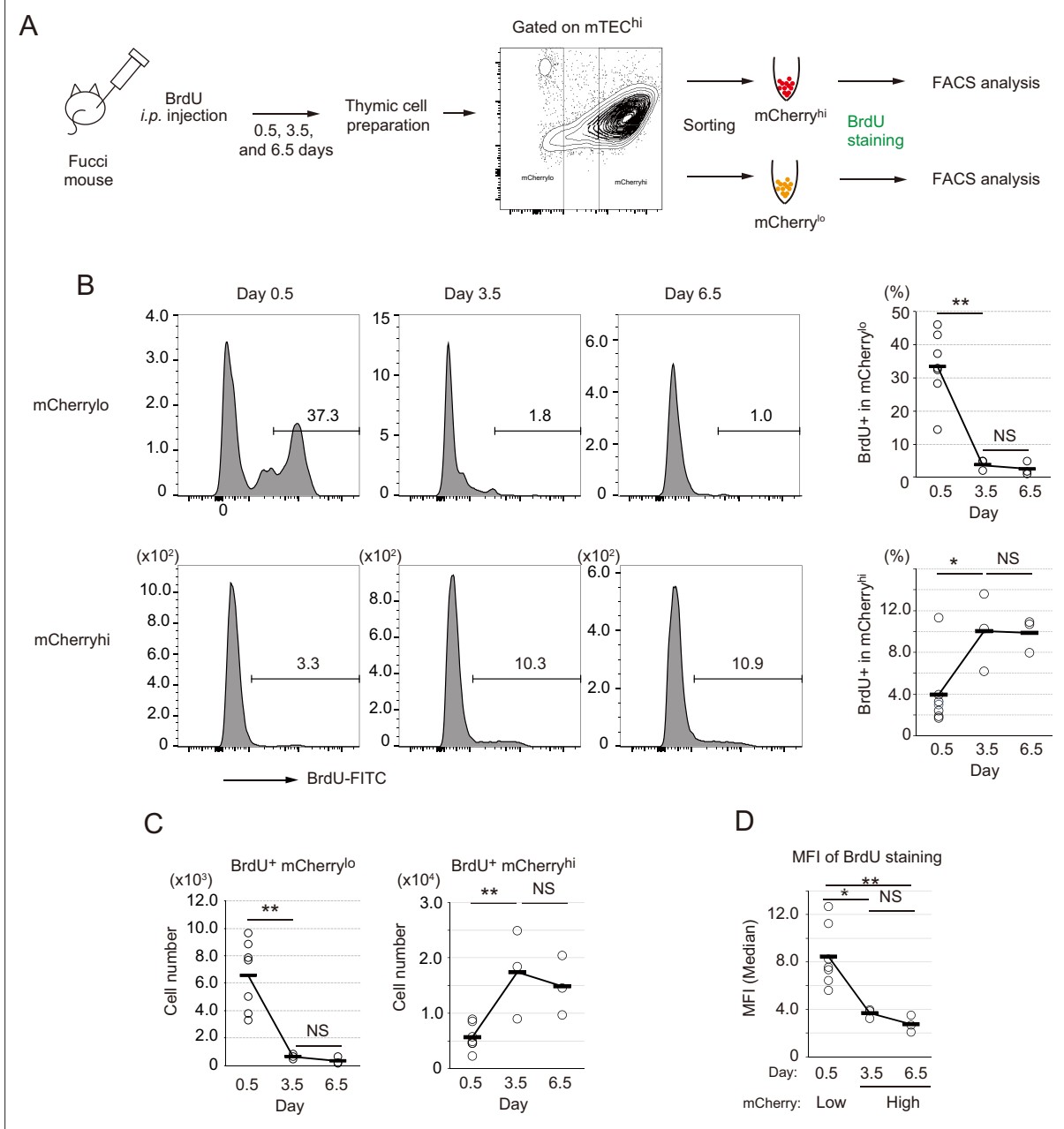

**Figure 6.** Fate mapping study with in vivo BrdU pulse-labeling of Fucci thymic epithelial cells (TECs). (**A**) Schematic procedure of in vivo BrdU pulse labeling of Fucci mice, and analysis of BrdU staining in mCherry$^{hi}$CD80$^{hi}$ and mCherry$^{lo}$CD80$^{hi}$ medullary TECs (mTECs) by flow cytometiric analysis. BrdU staining was done after sorting each cell fraction. (**B**) Typical flow cytometric profile of BrdU staining in mCherry$^{lo}$CD80$^{hi}$ mTECs (upper panels) and mCherry$^{hi}$CD80$^{hi}$ mTECs (lower panels) at days 0.5, 3.5, and 6.5 after the BrdU injection. Data for the ratio of BrdU$^+$ cells in each mTEC fraction are summarized in right-hand figures. N = 7 for 0.5 day after the BrdU injection, N = 3 for 3.5 and 6.5 days after the injection. Two-tailed Student's t-tests. **p < 0.01 and *p < 0.05. NS, not significant (p > 0.05). p = 1.5 × 10$^{-3}$ for the upper figure and p = 0.033 for the lower figure. Original data were shown in *Figure 6—source data 1*. (**C**) Cell number of BrdU$^+$mCherry$^{lo}$CD80$^{hi}$ mTECs and BrdU$^+$mCherry$^{hi}$CD80$^{hi}$ mTECs at days 0.5, 3.5, and 6.5 after the BrdU injection. Two-tailed Student's t-tests. **p < 0.01. NS, not significant (p > 0.05). p = 4.3 × 10$^{-3}$ for the left figure and p = 5.1 × 10$^{-3}$ for the right figure. Original data were shown in *Figure 6—source data 1*. (**D**) Mean fluorescence intensity (MFI) of BrdU staining in mCherry$^{lo}$CD80$^{hi}$ at day 0.5 and mCherry$^{hi}$CD80$^{hi}$ at days 3.5 and 6.5. MFIs of other time points were difficult to evaluate because of very low cell numbers. Two-tailed Student's t-tests. *p = 0.015 and **p = 6.5 × 10$^{-3}$. NS, not significant (p > 0.05). Original data were shown in *Figure 6—source data 1*.

The online version of this article includes the following source data for figure 6:

**Source data 1.** Related to *Figure 6B, C and D*.

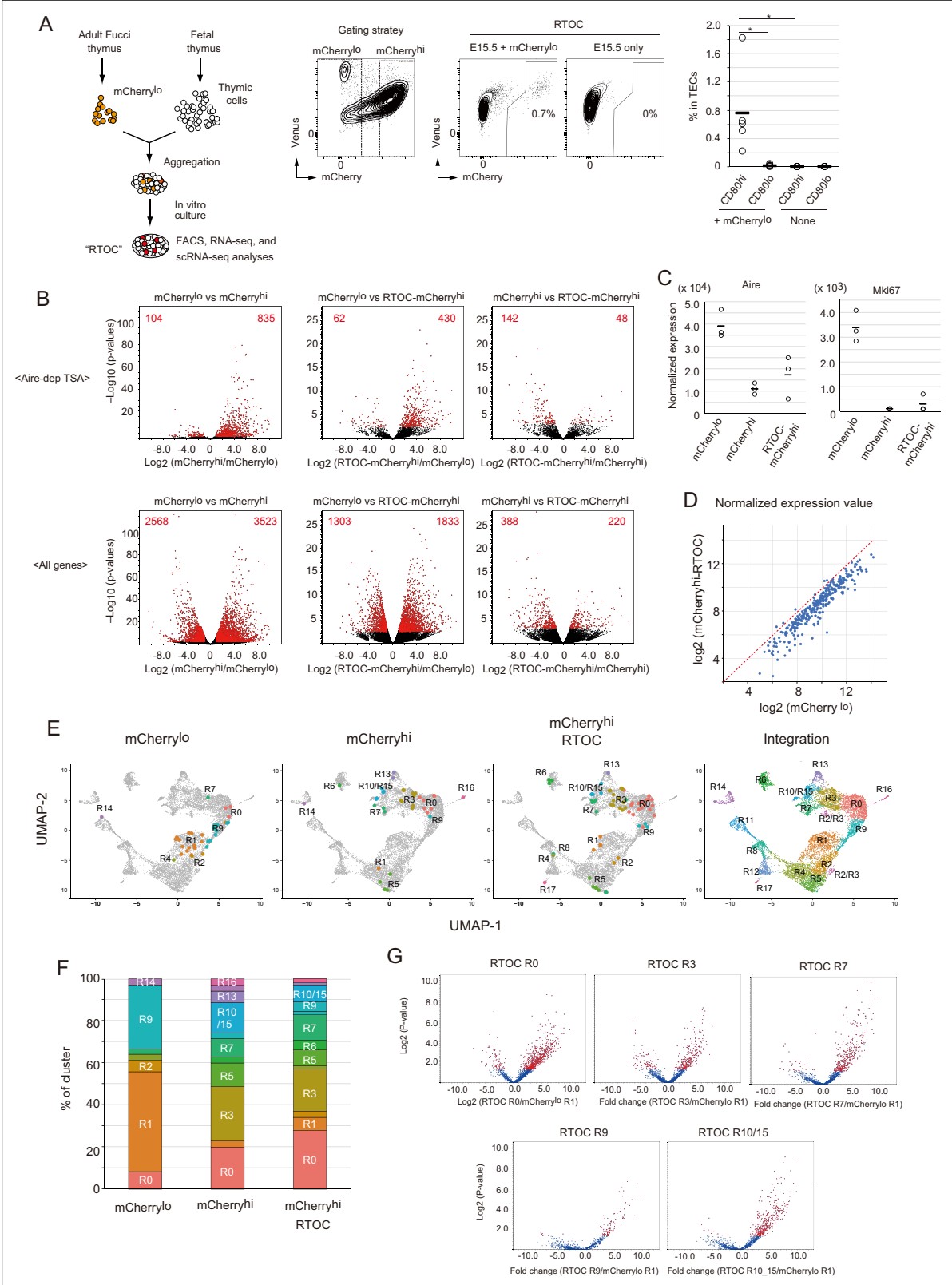

**Figure 7.** Fate mapping study of proliferating Aire[+] medullary thymic epithelial cells (mTECs) in in vitro reaggregated thymic organ culture (RTOC). (**A**) RTOC experiment to test the differentiation capacity of proliferating Aire[+] mTECs. Proliferating Aire[+] mTECs (mCherry[lo]) and E15.5 embryonic thymic cells were reaggregated and subsequently cultured for 5 days. Reaggregated thymic organ (RTO) was analyzed by flow cytometry. Representative flow cytometric profiles of RTOC are shown. N = 5. The ratio of mCherry[hi] cells in TECs is summarized in the right-hand figure. *p < 0.05. p = 0.027 between

*Figure 7 continued on next page*

CD80[hi] and CD80[lo] in mCherry[lo] and p = 0.024 between CD80[hi] mCherry[lo] and CD80[hi] RTOC control. (**B**) Volcano plots of RNA-seq data from mCherry[lo] CD80[hi] mTECs (mCherry[lo]), mCherry[hi] CD80[hi] mTECs (mCherry[hi]), and mCherry[hi] CD80[hi] mTECs in RTOC (mCherry[hi] in RTOC). Red dots in volcano plots indicate genes for which expression differed significantly between the two subsets. Numbers of differentially expressed genes are shown in the panels. N = 3. Y axis is log10 of FDR p-value. (**C**) Expression levels of Aire and Mki67 in mCherry[lo], mCherry[hi], and mCherry[hi] in RTOC. (**D**) Scatter plot of normalized expression values of TA-TEC marker candidates in mCherry[lo] and mCherry[hi] in RTOC. TA-TEC marker candidate genes were selected from bulk RNA-seq data and scRNA-seq data in *Figure 7—figure supplement 2* (**E**) Integration of well-based single-cell random displacement amplification sequencing (scRamDA-seq) data (mCherry[lo], mCherry[hi], and mCherry[hi] in RTOC) with the droplet-based scRNA-seq data in *Figure 2*. (**F**) Frequency of each cell cluster in scRamDA-seq data of mCherry[lo], mCherry[hi], and mCherry[hi] -RTOC. (**G**) Volcano plot of tissue-specific antigen (TSA) expression in each cell cluster in scRamDa-seq data of mCherry[hi]-RTOC as compared to mCherry[lo]. Red dots indicate significantly changed TSA genes.

The online version of this article includes the following figure supplement(s) for figure 7:

**Figure supplement 1.** Flow cytometric analysis and gene set enrichment analysis of reaggregation thymic organ culture experiments.

**Figure supplement 2.** Analysis of differentially expressed tissue specific antigen (TSA) genes between mCherry[hi] thymic epithelial cells (TECs) in reaggregated thymic organ culture (RTOC) and mCherry[lo] TECs.

**Figure supplement 3.** Integration of droplet-based single-cell RNA sequencing and well-based single-cell random displacement amplification sequencing data.

## Discussion

With regard to mTEC differentiation in the adult thymus (*Figure 8B*), we hypothesize that Aire[+] TA-TECs were generated from their Aire-negative progenitors. Aire[+] TA-TECs (cluster 3) undergo cell division and then differentiate into quiescent Aire[+] mTECs (cluster 0) through a transition stage, which corresponds to cluster R9 in scRNA-seq data. This differentiation process is accompanied by a chromatin structure change. Post-mitotic Aire[+] mTECs begin high-level TSA expression, and further differentiate into post-Aire mTECs (R7, R10, and R13) by closing the Aire enhancer region. Differentiation of mTECs expressing TSAs may have to coordinate differentiation with cell cycle regulation, as proposed in neural cells and muscle differentiation (*Ruijtenberg and van den Heuvel, 2016*).

Previous scRNA-seq of TEC and thymic stroma suggested the presence of a TEC subset expressing proliferative marker genes (*Dhalla et al., 2020*; *Baran-Gale et al., 2020*; *Wells et al., 2020*). Wells et al. proposed that TA-TECs belong to MHC class II (MHC II)[lo]Aire[−] subsets (*Wells et al., 2020*). In contrast, our analysis of Fucci and Aire reporter mice and cell fate mapping indicated the presence of CD80[hi] AIRE[+] TA-TECs. This inconsistency may be due to the fact that surface expression of MHC class II and CD80 in TA-TECs could be slightly lower than mature mTECs. Interestingly, analysis using the Fucci reporter suggested the existence of CD80[lo] Aire[−] proliferating mTECs. It is important to address whether Aire[−] proliferating TECs are TA-TECs or a proliferative population of CD80[lo] mTECs in the future.

Interestingly, although TA-TECs express AIRE protein, the expression level of Aire-dependent TSAs is much lower than that of quiescent AIRE[+] TECs. This suggests that AIRE protein is required, but not sufficient for induction of this TSA expression. Several scenarios may explain this fact. First, AIRE requires other regulators for TSA expression. It was reported that AIRE binds to various proteins, possibly regulators of AIRE-dependent TSA expression. Some of these essential regulators may be absent in Aire[+] TA-TECs. Another possibility is that cell proliferation inhibits TSA expression induced by AIRE mTECs. There may be mechanisms that suppress the AIRE function in TA-TECs, and thereby inhibit incidence of unfavorable cell states, such as tumor onset due to promiscuous gene expression.

Generally, in other tissues, TACs constitute a link between stem cells and mature cells (*Zhang and Hsu, 2017*). An important question is, 'What cells differentiate into proliferating Aire[+] mTECs?' Previous studies have suggested that mTECs[lo] expressing low levels of maturation markers, that is, CD80 or MHC II, are precursors (*Abramson and Anderson, 2017*; *Gray et al., 2007*). However, several recent studies have suggested that mTEC[lo] contains several subsets, including CCL21a-positive mTECs, tuft-like mTECs, and others. One possible explanation for this is that a small number of mTEC stem cells or other precursor cells may be present in the mTEC[lo] subset (*Gray et al., 2007*). Consistently, RNA velocity analysis also suggested that most mTEC[lo] cells do not appear to differentiate into Aire-expressing mTECs. Given that transit-amplifying mTECs are present, a small number of stem/precursor cells would theoretically be sufficient for mTEC reconstitution. A previous study proposed that TECs expressing claudin 3/4 and SSEA-1 had characteristic features of mTEC stem cells in embryonic thymus (*Sekai et al., 2014*). We failed to detect a corresponding cluster of mTEC

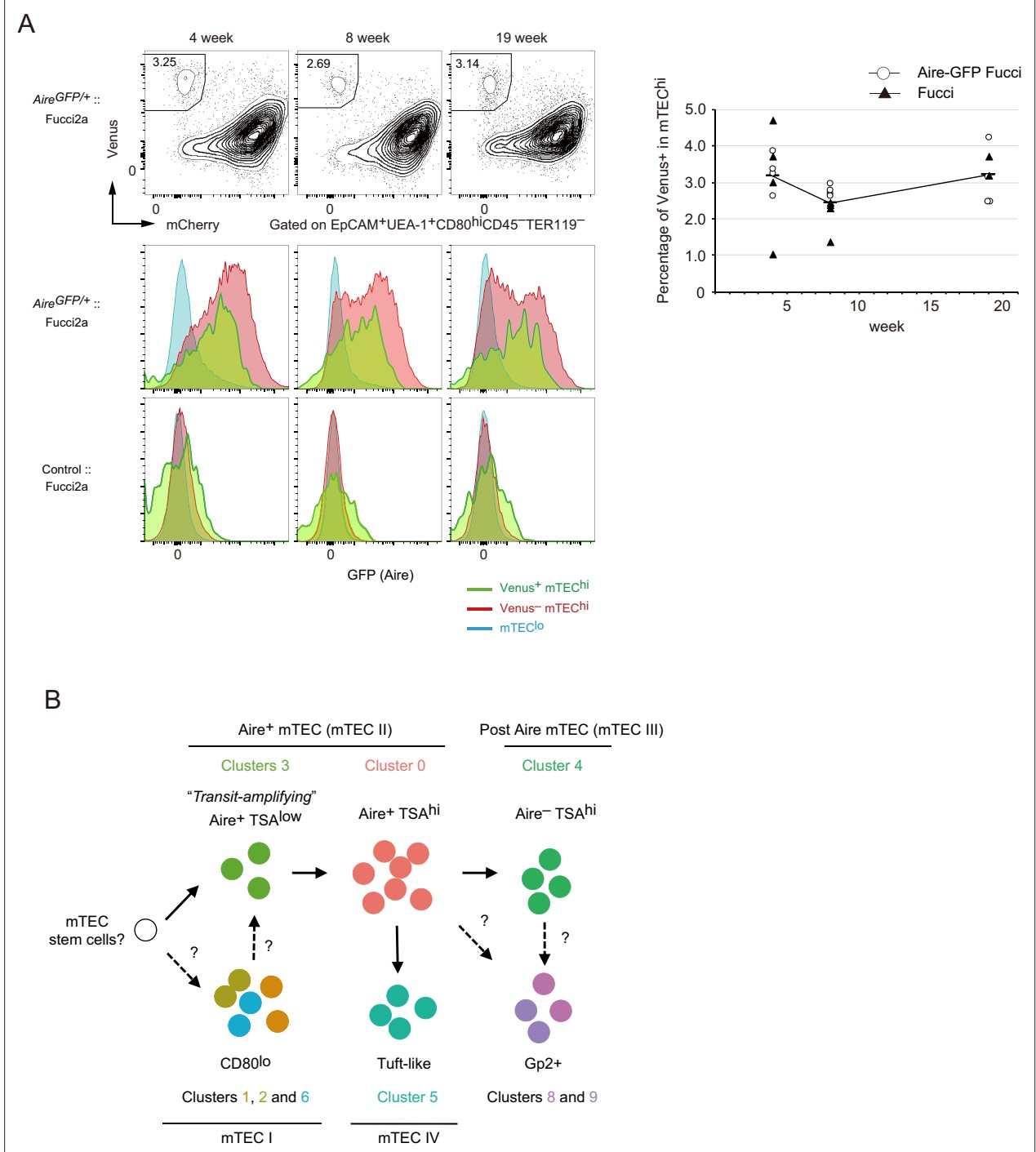

**Figure 8.** Proliferating Aire[+] CD80[hi] medullary thymic epithelial cells (mTECs) persist in older mice. (**A**) Flow cytometry analysis of CD80[hi] mTEC subsets from Fucci2a mice aged 4, 8, and 19 weeks. Representative data are shown. Percentages of Venus[+] cells in CD80[hi] mTEC subsets are summarized in the graph in the right panel. N = 4 each for *Aire*[gfp/+]:: Fucci2a (circle) and control::Fucci2a (closed triangles). (**B**) Schematic depiction of the proposed process of Aire[+] mTEC development in the adult thymus. Transit-amplifying TSA[lo] Aire[+] TECs give rise to mature mTECs. Precursor cells to the transit-amplifying TECs have not been determined yet. Cluster numbers in *Figure 1* are shown together with the model of mTEC subsets I to IV.

The online version of this article includes the following figure supplement(s) for figure 8:

**Figure supplement 1.** Integration of single-cell RNA sequencing data of 4-week-old and fetal thymic epithelial cell.

**Figure supplement 2.** Single-cell RNA sequencing (scRNA-seq) data of feta thymus (*Kernfeld et al., 2018*) were integrated with other adult scRNA-seq data (*Bornstein et al., 2018*; *Dhalla et al., 2020*; *Wells et al., 2020*).

stem cells as a subset of adult scRNA clusters. This may be because corresponding mTEC stem cells in adult thymus are included in the 'intertypical' TEC cluster, which may be a mixture of various TECs (*Baran-Gale et al., 2020*). More detailed characterization of mTEC stem cells in the adult thymus is necessary to illuminate differentiation dynamics of mTECs.

In this study, we performed and combined two experiments in scRNA-seq and scATAC-seq of TECs prepared from age- and gender-matched mice. Integration and splitting analysis of datasets suggested substantial reproducibility in both single-cell analyses. Especially, UMAP and clustering of scRNA-seq data were similar between the two experiments, even without batch effect reduction (*Figure 2—figure supplement 2*). This good consistency in cell clustering may be due to the multiple dimensions of single-cell data. However, it should be noted that only two droplet-based, single-cell analyses may be less reliable in terms of differential gene expression, because only a few thousand genes are normally detected in individual cells. Thus, it may be important to confirm the results obtained from single-cell analysis by using other ways, such as cytometric analysis.

Overall, the scRNA-seq analysis in the present study suggested the presence of a novel differentiation process of TECs in adult thymus. Disturbance of adult TEC homeostasis may cause thymoma, autoimmunity, and other diseases. Further characterization of molecular mechanisms underlying differentiation and maintenance processes in TECs will aid development of novel therapeutic strategies against these thymus-related diseases.

## Materials and methods
### Mice
C57BL/6 mice were purchased from Clea Japan. Littermates or age-matched, wild-type mice from the same colonies as the mutant mice were used as controls. *Aire-GFP* mice (CDB0479K, http://www2.brc.riken.jp/lab/animal/detail.php?brc_no=03515) and B6;129-Gt(ROSA)26Sor < tm1(Fucci2aR)Jkn> (RBRC06511) (Fucci2a) (*Mort et al., 2014*) were provided by the RIKEN BRC through the National Bio-Resource Project of the MEXT, in Japan. CAG-Cre transgenic mice were kindly provided by Dr Jun-ichi Miyazaki (*Sakai and Miyazaki, 1997*). B6(Cg)-Foxn1tm3(cre)Nrm/J are from Jackson Laboratory (*Gordon et al., 2007*). Fucci2a mice were crossed with CAG-Cre or Foxn1-Cre mice to activate mCherry and Venus expression. Fucci mice crossed with CAG-Cre were used for all experiments except for immunostaining experiments (*Figure 5*). All mice were maintained under specific pathogen-free conditions and handled in accordance with Guidelines of the Institutional Animal Care and Use Committee of RIKEN, Yokohama Branch (2018-075). Almost all of available mutant and control mice were randomly used for experiments without any selection.

### Preparation of TEC suspensions and flow cytometry analysis
Murine thymi were minced using razor blades. Thymic fragments were then pipetted up and down to remove lymphocytes. Then, fragments were digested in RPMI 1640 medium containing Liberase (Roche, 0.05 U/mL) plus DNase I (Sigma-Aldrich) via incubation at 37°C for 12 min three times. Single-cell suspensions were stained with anti-mouse antibodies. Dead cells were excluded via 7-aminoactinomycin D staining. Cells were sorted using a FACS Aria instrument (BD). Post-sorted cell subsets were determined to contain >95% of relevant cell types. Data were analyzed using Flowjo 10. No data points or mice were excluded from the study. Randomization and blinding were not used.

### Droplet-based scRNA-seq analysis
For scRNA-seq analysis, cell suspensions of thymi from three mice were prepared and pooled for each individual scRNA-seq experiment. Two experiments were performed. Cellular suspensions were loaded onto a Chromium instrument (10× Genomics) to generate a single-cell emulsion. scRNA-seq libraries were prepared using Chromium Single Cell 3′ Reagent Kits v2 Chemistry and sequenced in multiplex on the Illumina HiSeq2000 platform (rapid mode). FASTQ files were processed using Fastp (*Chen et al., 2018*). Reads were demultiplexed and mapped to the mm10 reference genome using Cell Ranger (v3.0.0). Processing of data with the Cell Ranger pipeline was performed using the HOKUSAI supercomputer at RIKEN and the NIG supercomputer at ROIS National Institute of Genetics. Expression count matrices were prepared by counting unique molecule identifiers. Downstream single-cell analyses (integration of two datasets, correction of dataset-specific batch effects,

UMAP dimension reduction, cell cluster identification, conserved marker identification, and regressing out cell cycle genes) were performed using Seurat (*Butler et al., 2018*). Briefly, cells that contained a percentage of mitochondrial transcripts > 15% were filtered out. Genes that were expressed in more than five cells and cells expressing at least 200 genes were selected for analysis. Two scRNA-seq datasets were integrated with a combination of Find Integration Anchors and Integrate Data functions (*Stuart et al., 2019*). Resolution was set as 0.6 for the FindCluster function. In subclustering analysis of cluster R1, the resolution of FindCluster function was set to 0.7. Murine cell cycle genes equivalent to human cell cycle genes listed in Seurat were used for assigning cell cycle scores.

For comparison with a previously reported RNA-seq dataset obtained via a well-based study (*Bornstein et al., 2018*), the expression matrix of unique molecule identifiers was used. Integration of the two datasets was performed using the Seurat package as described above. RNA velocity analysis was performed using velocyto. Bam/sam files obtained from the Cell Ranger pipeline were transformed to loom format on velocyto.py. RNA velocity was estimated and visualized using loom files by the velocyto R package and pagoda2.

## Droplet-based scATAC-seq analysis

In scRNA-seq analysis, cell suspensions of thymi from two mice were prepared and pooled for each individual scRNA-seq experiment. The EpCAM$^+$CD45$^-$TER119$^-$ fraction was collected using a cell sorter (BD Aria). After washing with PBS containing 0.04% BSA, sorted cells were suspended in lysis buffer containing 10 mM Tris-HCl (pH 7.4), 10 mM NaCl, 3 mM MgCl$_2$, 0.1% Tween-20, 0.1% NP-40, 0.01% Digitonin, and 1% BSA on ice for 3 min. Wash buffer containing 10 mM Tris-HCl (pH 7.4), 10 mM NaCl, 3 mM MgCl$_2$, 0.1% Tween-20, and 1% BSA was added to the lysed cells. After centrifuging the solution, a nuclear pellet was obtained by removing the supernatant and the pellet was re-suspended in wash buffer. The concentration of nuclei and their viability were determined by staining with acridine orange/propidium iodide, and 10,000 nuclear suspensions were loaded onto a Chromium instrument (10× Genomics) to generate a single-cell emulsion. scATAC-seq libraries were prepared using Chromium Next GEM Single-Cell ATAC Reagent Kits v1.1 and sequenced in multiplex on an Illumina HiSeq X Ten platform. Reads were demultiplexed and mapped to the mm10 reference genome with Cell Ranger ATAC. Processing data with the Cell Ranger pipeline was performed using the NIG supercomputer at ROIS National Institute of Genetics. Downstream analyses (integration of two datasets, correction of dataset-specific batch effects, UMAP dimension reduction, cluster identification, and identification of differentially accessible regions) were performed using Signac (v1.6) and the Harmony algorithm. Dimension reduction was done using latent semantic indexing. Cells were filtered according to the following parameters: peak_region_fragments 2000–20,000, percentage of fragments in peaks > 50%, blacklist_ratio < 0.03, nucleosome_signal < 0.8, and TSS enrichment > 2. To create a gene activity matrix from scATAC-seq data, the number of fragments in gene coordinates and their 2 kb upstream regions were counted. Integration of scATAC-seq and scRNA-seq data was performed with Signac (v1.6.) using the gene activity matrix in scATAC-seq. The gene activity matrix in scATAC-seq was transferred to scRNA-seq data.

## Well-based scRNA-seq analysis

Single cells were sorted into PCR tubes containing 1 μL of cell lysis solution (1:10 Cell Lysis buffer [Roche], 10 U/μL Rnasin plus Ribonuclease inhibitor [Promega]) using a cell sorter, shaken at 1400 rpm for 1 min with a thermo mixer, and then stored at –80°C. Cell lysates were denatured at 70°C for 90 s and held at 4°C until the next step. To eliminate genomic DNA contamination, 1 μL of genomic DNA digestion mix (0.5 × PrimeScript Buffer, 0.2 U of DNase I Amplification Grade, in RNase-free water) was added to 1 μL of the denatured sample. The mixtures were mixed by gentle tapping, incubated in a T100 thermal cycler at 30°C for 5 min and held at 4°C until the next step. One microliter of RT-RamDA mix (2.5× PrimeScript Buffer, 0.6 pmol oligo(dT)18, 8 pmol 1st-NSRs, 100 ng of T4 gene 32 protein, and 3× PrimeScript enzyme mix in RNase-free water) was added to 2 μL of the digested lysates. The mixtures were mixed with gentle tapping, and incubated at 25°C for 10 min, 30°C for 10 min, 37°C for 30 min, 50°C for 5 min, and 94°C for 5 min. Then, the mixtures were held at 4°C until the next step. After RT, the samples were added to 2 μL of second-strand synthesis mix containing 2.25 × NEB buffer 2 (NEB), 0.625 mM each dNTP Mixture (NEB), 40 pmol 2nd-NSRs, and 0.75 U of Klenow Fragment (NEB) in RNase-free water. Mixtures were again mixed by gentle tapping, and incubated

at 16°C for 60 min, 70°C 10 min and then at 4°C until the next step. The above-described double-stranded cDNA was purified using 15 μL of AMPure XP SPRI beads (Beckman Coulter) diluted twofold with Pooling buffer (20% PEG8000, 2.5 M NaCl, 10 mM Tris-HCl pH 8.0, 1 mM EDTA, 0.01% NP40) and Magna Stand (Nippon Genetics). Washed AMPure XP beads attached to double-stranded cDNAs were directly eluted using 3.75 μL of 1× Tagment DNA Buffer (10 mM Tris-HCl pH 8.5, 5 mM MgCl$_2$, 10% DMF) and mixed well using a vortex mixer and pipetting. Diluted Tn5-linker complex was added to the eluate and the mixture was incubated at 55°C for 10 min, then 1.25 μL of 0.2% SDS was added and incubated at room temperature for 5 min. After PCR for adaptor ligation, sequencing library DNA was purified using 1.0× the volume of AMPure XP beads and eluted into 24 μL of 10 mM Tris-Cl, pH 8.5. Reads were demultiplexed and mapped to the mm10 reference genome with STAR. Cells with detected read counts less than half and greater than 1.8 times of the average count were omitted from the analysis. Fucci-negative cells were also removed. Integration of well-based scRNA-seq data with 10× scRNA-seq data and UMAP dimension were performed using Seurat (*Butler et al., 2018*). Genes that were expressed in more than five cells and cells expressing at least 200 genes were selected for analysis.

## Standard RNA-seq analysis

Total RNA was prepared using TRIzol reagent (Thermo Fisher Scientific) in accordance with the manufacturer's protocol. After rRNA was depleted using the NEBNext rRNA Depletion Kit, the RNA sequence library was prepared using the NEBNext Ultra Directional RNA Library Prep Kit (New England Biolabs). Paired-end sequencing was performed with NextSeq500 (Illumina). Sequence reads were quantified for annotated genes using CLC Genomics Workbench (Version 7.5.1; Qiagen). Gene expression values were cut off at a normalization expression threshold value of 3. Differential expression was assessed via empirical analysis with the DGE (edgeR test) tool in CLC Main Workbench, in which the Exact Test of Robinson and Smyth was used (*Robinson and Smyth, 2008*). An FDR-corrected p value was used for testing statistics for RNA-seq analysis. Previously described lists of TSAs and Aire-dependent TSAs (*Sansom et al., 2014*) were used for the analysis.

## RTOC and RNA-seq analysis

mCherry$^{lo}$ cells ($4 × 10^4$–$1 × 10^5$) were sorted from Fucci mice and subsequently reaggregated with trypsin-digested thymic cells ($1$–$2 × 10^6$) from E15.5 wild-type mice. RTOCs were cultured on Nucleopore filters (Whatman) placed in R10 medium containing RPMI1640 (Wako) supplemented with 10% fetal bovine serum, 2 mM L-glutamine (Wako), 1× nonessential amino acids (Sigma-Aldrich), 0.1 pM cholera Toxin Solution (Wako 030-20621), 5 μg/mL insulin solution from bovine pancreas (SIGMA I0516-5ML), 2 nM triiodo-L-thyronine (SIGMA T2877-100MG), 1000 units/mL LIF (nacalai NU0012-1), 0.4 μg/mL hydrocortisone,10 ng/mL EGF (Gibco PMG8041), 1 μg/mL RANKL (Wako), penicillin-streptmycin mixed solution (Nacalai Tesque), and 50 μM 2-mercaptoethanol (Nacalai Tesque) for 5 days. For RNA-seq of RTOC experiments, RamDA-seq was used (*Hayashi et al., 2018*), which allows RNA-seq analysis of low numbers of cells. Briefly, sorted cells were lysed in TCL buffer (Qiagen). After purification of nucleic acids with Agencourt RNA Clean XP (Beckman Coulter) and subsequent treatment with DNase I, the RT-RamDA mixture containing 2.5× PrimeScript Buffer (TAKARA), 0.6 μM oligo(dT)18 (Thermo), 10 μM 1st NSR primer mix, 100 μg/mL of T4 gene 32 protein, and 3× PrimeScript enzyme mix (TAKARA) were added to the purified nucleic acids for reverse transcription. Samples were added to second-strand synthesis mix containing 2× NEB buffer 2 (NEB), 625 nM dNTP Mixture (NEB), 25 μM 2nd NSR primers, and 375 U/mL of Klenow Fragment (3'–5' exo-) (NEB). After cDNA synthesis and subsequent purification by AMPure XP (Beckman Coulter), sequencing library DNA was prepared using the Tn5 tagmentation-based method. Single-read sequencing was performed using a HiSeq2500 (v4, high out mode). Sequence reads were quantified for annotated genes using CLC Genomics Workbench (Version 7.5.1; Qiagen).

## Immunohistochemistry

The thymus was fixed with 4% paraformaldehyde and frozen in OCT compound. After washing cryosections (5 μm) with PBS, sections were blocked with 10% normal goat serum. Keratin-5 was detected using a combination of a polyclonal rabbit anti-mouse keratin-5 antibody (1:500) and Alexa Fluor

647-donkey-anti-rabbit IgG. Aire was detected using a labeled monoclonal antibody (1:300). Confocal color images were obtained using an LAS X (Leica) microscope.

## Immunocytochemistry

Thymic cell suspensions prepared via Liberase digestion were stained with anti-CD45-PE and anti-TER119-PE. After depletion of labeled CD45$^+$ and TER119$^+$ cells via anti-PE microbeads and a magnetic-activated cell sorting separator, negatively selected cells were stained with antiEpCAM (CD326), anti-CD80, anti-Ly51, and UEA-1. Venus$^+$ CD80$^{hi}$ mTECs were sorted and spun down on glass slides using a cytospin. Slides were then fixed with acetone and stained with anti-Aire antibody and DAPI for nuclear staining. Confocal images were obtained using an LAS X microscope.

## Statistical analysis

Statistically significant differences between mean values were determined using Student's t-test (***p < 0.001, **p < 0.01, and *p < 0.05). Principle component analysis was performed using the prcomp function in R-project. The sample size was not predetermined by statistical methods, but was based on common practice and previous studies (*Akiyama et al., 2016*; *Akiyama et al., 2014*). All replicates are biological replicates. All outliers were included in the data.

## Acknowledgements

The authors declare no competing financial interests. We thank the sequencing staff at the RIKEN Center for Integrative Medical Sciences for assisting with RNA-seq. Computations were performed on the NIG supercomputer at ROIS, National Institute of Genetics and HOKUSAI supercomputer at ISD, RIKEN.

---

## Additional information

### Funding

| Funder | Grant reference number | Author |
| --- | --- | --- |
| Japan Society for the Promotion of Science | Grants-in-Aid for Scientific Research 17H04038 | Taishin Akiyama |
| Japan Society for the Promotion of Science | Grants-in-Aid for Scientific Research 20H03441 | Taishin Akiyama |
| Japan Society for the Promotion of Science | Grants-in-Aid for Scientific Research 17K08622 | Nobuko Akiyama |
| Japan Society for the Promotion of Science | Grants-in-Aid for Scientific Research 20K07332 | Nobuko Akiyama |
| Princess Takamatsu Cancer Research Fund | | Taishin Akiyama |
| Uehara Memorial Foundation | | Taishin Akiyama |
| NOVARTIS Foundation | | Taishin Akiyama |
| Ministry of Education, Culture, Sports, Science and Technology | 18H04989 | Taishin Akiyama |
| Ministry of Education, Culture, Sports, Science and Technology | 19H04821 | Nobuko Akiyama |
| Japan Science and Technology Agency | CREST JPMJCR2011 | Taishin Akiyama |

The funders had no role in study design, data collection and interpretation, or the decision to submit the work for publication.

## Author contributions
Takahisa Miyao, Data curation, Formal analysis, Investigation, Writing – original draft; Maki Miyauchi, Data curation, Formal analysis, Investigation, Validation; S Thomas Kelly, Tommy W Terooatea, Formal analysis, Investigation; Tatsuya Ishikawa, Data curation, Formal analysis, Investigation; Eugene Oh, Hiroto Ishii, Haruka Yabukami, Masafumi Muratani, Data curation; Sotaro Hirai, Kenta Horie, Houko Ohki, Yuya Maruyama, Takao Seki, Masaki Yoshida, Data curation, Formal analysis; Yuki Takakura, Aki Minoda, Formal analysis, Validation; Mio Hayama, Formal analysis, Investigation, Validation; Azusa Inoue, Atsushi Miyawaki, Methodology, Supervision, Validation; Asako Sakaue-Sawano, Methodology, Supervision; Nobuko Akiyama, Formal analysis, Funding acquisition, Investigation, Supervision, Writing – review and editing; Taishin Akiyama, Formal analysis, Funding acquisition, Investigation, Project administration, Supervision, Validation, Writing – original draft, Writing – review and editing

## Author ORCIDs
S Thomas Kelly http://orcid.org/0000-0003-3904-6690
Eugene Oh http://orcid.org/0000-0001-8534-0183
Masafumi Muratani http://orcid.org/0000-0002-0276-8000
Nobuko Akiyama http://orcid.org/0000-0003-1085-5996
Taishin Akiyama http://orcid.org/0000-0002-0341-6154

## Ethics
All mice were handled in accordance with Guidelines of the Institutional Animal Care and Use Committee of RIKEN, Yokohama Branch . Permission #2018-075.

## Decision letter and Author response
Decision letter https://doi.org/10.7554/eLife.73998.sa1
Author response https://doi.org/10.7554/eLife.73998.sa2

---

# Additional files

## Supplementary files
• Supplementary file 1. List of genes specifically expressed in each cluster.

• Supplementary file 2. Percentage of single-cell RNA sequencing (scRNA-seq) clusters in single-cell assays for transposase-accessible chromatin sequencing (scATAC-seq) clusters after the integration.

• Supplementary file 3. Differentially chromatin-accessible regions between cluster 0 and cluster 4.

• Supplementary file 4. Percentage of single-cell RNA sequencing (scRNA-seq) subclusters of R1 in single-cell assays for transposase-accessible chromatin sequencing (scATAC-seq) clusters after the integration.

• Supplementary file 5. Gene ontology (GO) analysis of genes differentially expressed in Venus[+] cells.

• Supplementary file 6. List of all, Aire-dependent, Aire-independent tissue-specific antigen genes.

• Supplementary file 7. Summary for assignment of individual single cells in single-cell random displacement amplification sequencing (scRamDa-seq) of mCherryhi, mCherrylo, and mCherryhi-RTOC.

• Supplementary file 8. Summary for assignment of individual single cells in single-cell random displacement amplification sequencing (scRamDa-seq) of mCherryhi, mCherrylo, and mCherryhi-RTOC.

• Transparent reporting form

## Data availability
FASTQ data of RNA-Seq and ATAC-seq are deposited in DDBJ (DRA009125, DRA010209 , DRA012308, DRA012309, DRA012452, and DRA013875).

The following datasets were generated:

| Author(s) | Year | Dataset title | Dataset URL | Database and Identifier |
|---|---|---|---|---|
| Akiyama T | 2019 | Single cell analysis of thymic epithelial cells | https://ddbj.nig.ac.jp/public/ddbj_database/dra/fastq/DRA009/DRA009125/ | DDBJ, DRA009125 |
| Akiyama T | 2020 | RNA-seq analysis of transit-amplifying Aire+ mTECs | https://ddbj.nig.ac.jp/public/ddbj_database/dra/fastq/DRA010/DRA010209 | DDBJ, DRA010209 |
| Akiyama T | 2021 | Single cell analysis of transit-amplifying thymic epithelial cells | https://ddbj.nig.ac.jp/public/ddbj_database/dra/fastq/DRA012/DRA012308/ | DDBJ, DRA012308 |
| Akiyama T | 2021 | RNA-seq analysis of transit amplifying mTEC in reaggregation thymic organ culture | https://ddbj.nig.ac.jp/public/ddbj_database/dra/fastq/DRA012/DRA012309/ | DDBJ, DRA012309 |
| Akiyama T | 2021 | Single cell ATAC sequence analysis of thymic epithelial cells (Exp.1) | https://ddbj.nig.ac.jp/public/ddbj_database/dra/fastq/DRA012/DRA012452/ | DDBJ, DRA012452 |
| Akiyama T | 2022 | Single cell ATAC sequence analysis of thymic epithelial cells (Exp.2) | https://ddbj.nig.ac.jp/public/ddbj_database/dra/fastq/DRA013/DRA013875/ | DDBJ, DRA013875 |

The following previously published datasets were used:

| Author(s) | Year | Dataset title | Dataset URL | Database and Identifier |
|---|---|---|---|---|
| Amit I, Abramson J, Bornstein C, Nevo S, Giladi A, Kadouri N | 2018 | Large-scale single cell mapping of the thymic stroma identifies a new thymic epithelial cell lineage | https://www.ncbi.nlm.nih.gov/geo/query/acc.cgi?acc=GSE103967 | NCBI Gene Expression Omnibus, GSE103967 |
| Dhalla F, Baran-Gale J, Maio S, Chappell L, Hollander G, Ponting CP | 2019 | Single cell RNA-seq of medullary thymic epithelial cells (mTEC) | https://www.ebi.ac.uk/arrayexpress/experiments/E-MTAB-8105/ | ArrayExpress, E-MTAB-8105 |
| Kernfeld E, Genga R, Magaletta M, Neherin K, Xu P, Maehr R | 2018 | Single-cell RNA sequencing resolves cellular heterogeneity throughout embryonic development of the thymus | https://www.ncbi.nlm.nih.gov/geo/query/acc.cgi?acc=GSE107910 | NCBI Gene Expression Omnibus, GSE107910 |
| Wells KL, Miller CN, Gschwind AR, Phipps JD, Anderson MS, Steinmetz LM | 2020 | Single cell sequencing defines a branched progenitor population of stable medullary thymic epithelial cells | https://www.ncbi.nlm.nih.gov/geo/query/acc.cgi?acc=GSE137699 | NCBI Gene Expression Omnibus, GSE137699 |

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

# Appendix 1

## Appendix 1—key resources table

| Reagent type (species) or resource | Designation | Source or reference | Identifiers | Additional information |
|---|---|---|---|---|
| Genetic reagent (*Mus musculus*) | B6.Cg-Aire < tm2Mmat>/Rbrc | RIKEN BioResource Research Center | BRC No:RBRC03515 | |
| Genetic reagent (*Mus. musculus*) | B6;129-Gt(ROSA)26Sor < tm1(Fucci2aR)Jkn> | RIKEN BioResource Research Center | BRC No:RBRC06511 | |
| Genetic reagent (*Mus. musculus*) | B6(Cg)-Foxn1tm3(cre)Nrm/J | Jackson Laboratory | IMSR Cat#JAX:018448, RRID:IMSR_JAX:018448 | |
| Genetic reagent (*Mus. musculus*) | CAG-Cre transgenic mice | Provided by Jun-ichi Miyazaki | | |
| Antibody | Purified anti-mouse CD16/32(Rat monoclonal) | BioLegend | Cat#101302, RRID:AB_312801 | FACS(1:200) |
| Antibody | APC/Cyanine7 anti-mouse CD45(Rat monoclonal) | BioLegend | Cat#103116, RRID:AB_312981 | FACS (1:200) |
| Antibody | PE Rat anti-mouse CD45(Rat monoclonal) | eBioscience | Cat#12-0451-82, RRID:AB_465668 | FACS (1:200) |
| Antibody | APC/Cyanine7 anti-mouse TER-119/Erythroid Cells(Rat monoclonal) | BioLegend | Cat#116223, RRID:AB_2137788 | FACS (1:200) |
| Antibody | PE anti-mouse TER-119/Erythroid Cells(Rat monoclonal) | eBioscience | Cat#12-5921-82, RRID:AB_466042 | FACS (1:200) |
| Antibody | Brilliant Violet 510 anti-mouse CD326 (Ep-CAM)(Rat monoclonal) | BioLegend | Cat#118231, RRID:AB_2632774 | FACS (1:400) |
| Antibody | FITC anti-mouse CD326 Ep-CAM (Rat monoclonal) | BioLegend | Cat#118208, RRID:AB_1134107 | FACS (1:400) |
| Antibody | Alexa Fluor 647 anti-mouse Ly-51 (Rat monoclonal) | BioLegend | Cat#108312, RRID:AB_2099613 | FACS (1:400) |
| Chemical compound, drug | Biotinylated Ulex Europaeus Agglutinin I (UEA I) | Vector Laboratories | Cat#B-1065–2 | FACS (1:800) |
| Chemical compound, drug | Streptavidin PE/Cyanine7 Conjugate | eBioscience | Cat#25-4317-82 | FACS (1:800) |
| Chemical compound, drug | Streptavidin APC/Cyanine7 Conjugate | BD Pharmingen | Cat#554063 RRID:AB_10054651 | FACS (1:400) |
| Antibody | PE anti-mouse CD80 (Armenian hamster monoclonal) | eBioscience | Cat#12-0801-81, RRID:AB_465751 | FACS (1:300) |
| Antibody | Pacific Blue anti-mouse CD80 Antibody (Armenian hamster monoclonal) | BioLegend | Cat#104724, RRID:AB_2075999 | FACS (1:300) |
| Antibody | Alexa Fluor 647 anti-mouse Aire (Rat monoclonal) | eBioscience | Cat#51-5934-80 | IHC (1:100) |
| Antibody | Purified Rabbit anti-Keratin 5 (rabbit polyclonal) | BioLegend | Cat#905504, RRID:AB_2616956 | IHC (1:400) |
| Antibody | Alexa Fluor 647 anti-Rabbit IgG (H + L) (Donkey polyclonal) | Invitrogen | Cat#A-31573, RRID:AB_2536183 | IHC (1:1000) |
| Chemical compound, drug | Liberase TM | Roche Diagnostics | Cat#5401127001 | |
| Chemical compound, drug | 7-Aminoactinomycin D | Calbiochem | Cat#129935-1MGCN | |
| Chemical compound, drug | SYTOX Blue Nucleic Acid Stain | Invitrogen | Cat#S11348 | |
| Software, algorithm | FlowJo version 10 | BD | FlowJo, RRID:SCR_008520 | |
| Software, algorithm | Cell Ranger v3.0.0 | 10× Genomics | Cell Ranger, RRID:SCR_017344 | |
| Software, algorithm | SEURAT version 4.1.0 | https://github.com/satijalab/seurat/blob/master/vignettes/install.Rmd | SEURAT, RRID:SCR_007322 | |

*Appendix 1 Continued on next page*

*Appendix 1 Continued*

| Reagent type (species) or resource | Designation | Source or reference | Identifiers | Additional information |
|---|---|---|---|---|
| Software, algorithm | Velocyto version 0.6 | https://github.com/velocyto-team/velocyto.R | Velocyto, RRID:SCR_018167 | |
| Software, algorithm | pagoda2 version 1.0.9 | https://github.com/kharchenkolab/pagoda2 | pagoda2, RRID:SCR_017094 | |
| Software, algorithm | Cell Ranger ATAC version1.1.0 | 10× Genomics | Cell Ranger ATAC, RRID:SCR_021160 | |
| Software, algorithm | Signac version 1.5.0 | https://github.com/timoast/signac/blob/master/vignettes/install.Rmd | Signac, RRID:SCR_021158 | |
| Software, algorithm | Monocle3, version 0.2.3 | https://cole-trapnell-lab.github.io/monocle3/docs/installation/ | Monocle3, RRID:SCR_018685 | |
| Software, algorithm | CLC Genomics Workbench Version 7.5.1 | QIAGEN | CLC Genomics Workbench, RRID:SCR_011853 | |
| Commercial assay or kit | Chromium Single Cell 3' Library & Gel Bead Kit v2 | 10× Genomics | Cat#PN-120237 | |
| Commercial assay or kit | Chromium Single Cell A Chip Kit | 10× Genomics | Cat#PN-120236 | |
| Commercial assay or kit | Chromium i7 Multiplex Kit | 10× Genomics | Cat#PN-120262 | |
| Commercial assay or kit | Chromium Next GEM Single Cell ATAC Library & Gel Bead Kit | 10× Genomics | Cat#PN-1000176 | |
| Commercial assay or kit | Chromium Next GEM Chip H Single Cell Kit | 10× Genomics | Cat#PN-1000162 | |
| Commercial assay or kit | Single Index Kit N, Set A | 10× Genomics | Cat#PN-1000212 | |
| Commercial assay or kit | NEBNext rRNA Depletion Kit | New England Biolabs | Cat#E6310 | |
| Commercial assay or kit | NEBNext Ultra Directional RNA Library Prep Kit for Illumina | New England Biolabs | Cat#E7420 | |
| Commercial assay or kit | KAPALibraryQuantificationKits Illumina/Universal | Nippon Genetics | Cat#KK4824 | |
| Chemical compound, drug | KAPAHiFi DNA Polymerase | Nippon Genetics | Cat#KK2102 | |
| Commercial assay or kit | Agilent High Sensitivity DNA Kit | Agilent Technologies | Cat#5067–4626 | |
| Commercial assay or kit | Multina DNA-12000 | SHIMADZU | Cat#S292-36600-91 | |

