## [Editor Report]

This report shows by scRNAs-seq and scATAC-seq the presence of a population of proliferating medullary thymic epithelial cells (mTECs) with a specific chromatin structure and high expression of Aire and CD80. Such Aire-expressing transit-amplifying mTECs may play a key role in establishing immunological self-tolerance.

---

## [Decision Letter]

**Decision letter after peer review:**

Thank you for submitting your article "Integrative analysis of scRNAs-seq and scATAC-seq revealed transit-amplifying thymic epithelial cells expressing autoimmune regulator" for consideration by *eLife*. Your article has been reviewed by 3 peer reviewers, one of whom is a member of our Board of Reviewing Editors, and the evaluation has been overseen by Betty Diamond as the Senior Editor. The reviewers have opted to remain anonymous.

Essential revisions:

1. The scATACseq experiment included a relatively low number (n=2) of mice. Understandably this is a complicated experiment, however, the variation between individual animals remains unknown and may influence the interpretation of the results. The authors are advised to show the clustering analysis for the two animals separately in a supplement to confirm that the changes are seen in both animals, not just in one. Similarly, the individual scRNA-seq UMAPs of the three animals should be included in the supplement. The authors should discuss the limitations of small groups in single-cell experiments in Discussion. Technically the integration of scATAC and scRNA seq results may be feasible.

2. Genes expressed in clusters should be provided in the supplement. Without this information, it is difficult to see the feasibility of the cluster annotations and to compare them with other similar studies. The authors should give an unbiased list of all genes expressed in mTECs, in particular those genes that are expressed in TACs. To harmonize the findings in the field, it would be useful for the readers if the authors would compare their cluster-specific genes for the overlap with TEC single-cell clusters from other studies (Bornstein et al., 2018, Dhalla et al., 2020, Baran-Gale et al., 2020, Wells et al., 2020). In addition, Wells et al., 2020 reported mTEC TACs to be mTEClo preceding Aire expression and giving rise to both Aire+ and Ccl21a+ cells. How do the authors reconcile their results (TAC as Aire+) with Wells et al., paper (TACs as Aire-)? How do they interpret the finding that Wells et al., find Ki67 significantly higher in mTEClo than in mTEChi, implying that cell division precedes high the expression of Aire?

4. The candidate cell population for TACs, cluster R1 expresses Aire and the proliferating cell marker Mki67. R1 also expresses Ccl21a. The authors did subclustering of R1 and found that R1A-D express Aire whereas R1E has a higher expression of Ccl21a. The authors note that "Thus, it is possible that TECs expressing cell-cycle-related genes, proposed by scRNA-seq analysis, contain at least two proliferating TECs subsets having different chromatin accessibilities and gene expression profiles." To confirm that both R1A-D and R1E subsets are proliferating TACs, the authors might show the proliferating gene markers in these subsets. Was Mki67 expressed among all R1 subpopulations? Would this argue for the presence of TAC among both Aire+ and Aire- cell populations? Assuming that TACs as a proliferating cell type should express multiple genes associated with cell cycling, the authors focus on Mki67 only, but did R1 TAC express other proliferating cell markers which would support the claim that these are indeed actively dividing cells?

6. The authors focus their study on the CD80high cycling cells (Aire+). Figure 4 show transcription profiles of the isolated cycling CD80+ mTECs. Dots in the "TSA genes" panel (right) don't appear in the "All genes" panel (left). Are those lost dots of TSA genes?

In Figure 4E (left) low expressed genes seem to be skewed towards Venus- mTEChi (in comparison to high expressed genes). A statistical assessment for the comparisons of the TSA, Aire-dep TSA and Aire-indep TSA profiles to the general profile (Figure 4E and 4G), considering expression levels, would confirm the visual assessment.

They should also discuss the reason why m-cherry low cells express a lower level of tissue-specific antigens even though they express Aire? Is Aire expression alone insufficient for TSA expression? Does the transcriptome data provide any mechanistic insight? In Fig4F, Aire expression is similar between Venus+ and Venus-. Is it compatible with Figure 4A showing less Aire-expressing cells in Venus+ than in Venus-?

7. In Figure 6, they showed that RTOC culture of mCherry-low cells produced mCherry-high cells, thereby they claimed that mCherry-low cells differentiated into mCherry-high cells. To substantiate this notion, they should rule out the possibility of selective cell survival of possibly contaminated mCherry-high cells or that transferring adult mTECs into the embryonic cell environment may trigger other signaling pathways that induce the upregulation of the cell cycle and mCherry expression? For example, what about the expression profiles of cell-death/apoptosis-related genes in mCherry low and high cells? What about the cell number of mCherry high cells after RTOC culture using mCherry-high cells alone as a control group? Is the cellularity of survivors comparable to RTOC culture using mCherry-low as shown in figure 6A? Would the same happen if they would transfer these cells to the adult thymus?

8. The authors nicely confirm, by the fine analysis of their scRNA-seq data, that the TAC population contains Aire+ cells and Ccl21+ cells in a visually mutually exclusive manner. However, they don't formally clarify whether the Ccl21-expressing TACs have differences in their chromatin accessibility pattern compared to the Aire-expressing TACs.

It's also worth showing the expression of CD80 in the scRNA-seq UMAP of TACs alone and in the one of all TECs (Figure 2). This would notably allow to determine whether CD80 expression is restricted to Aire-positive TACs or encompasses Aire-negative TACs (Ccl21+).

Also, projection of the R1A-E scRNAseq clusters onto the scATAC-seq UMPA would be enlightening.

9. Trajectory analyses provide a nice confirmation of published results identifying a trajectory from TACs to mTEChi. However, the authors don't discuss whether their data support a potential trajectory from TACs to mTEClo (already suggested/ reported) which seems to be present in Figure 3-Figure sup 2B. Would this mean that some TACs could mature into Ccl21+ mTECs (mTEClo)? and if so, how Aire and Ccl21 are expressed in these TACs? Which TAC sub-cluster do they belong to?

10. In figure S7, the data of fetal thymi showed that proliferating fetal mTECs might have a gene expression profile different from the adult counterpart. One caveat of the interpretation of the data is that the difference between the public data and the authors' data might be attributed to a batch effect because the clusters from two data sets seemed almost completely discrete. They should mention how they processed the data to diminish the risk of such artifacts. Or, it is better to add the data of fetal thymi obtained by the authors themselves, if possible.

*Reviewer #1 (Recommendations for the authors):*

Regarding figure 4, they should discuss the reason why m-cherry low cells express a lower level of tissue-specific antigens even though they express Aire? Is Aire expression alone insufficient for TSA expression? Does the transcriptome data provide any mechanistic insight?

In figure 6, they showed that RTOC culture of mCherry-low cells produced mCherry-high cells, thereby they claimed that mCherry-low cells differentiated into mCherry-high cells. To substantiate this notion, they should rule out the possibility of selective cell survival of possibly contaminated mCherry-high cells. For example, what about the expression profiles of cell-death/apoptosis-related genes in mCherry low and high cells? What about the cell number of mCherry high cells after RTOC culture using mCherry-high cells alone as a control group? Is the cellularity of survivors comparable to RTOC culture using mCherry-low as shown in figure 6A?

In figure S7, the data of fetal thymi showed that proliferating fetal mTECs might have a gene expression profile different from the adult counterpart. One caveat of the interpretation of the data is that the difference between the public data and the authors' data might be attributed to a batch effect because the clusters from two data sets seemed almost completely discrete. They should mention how they processed the data to diminish the risk of such artifacts. Or, it is better to add the data of fetal thymi obtained by the authors themselves, if possible.

*Reviewer #2 (Recommendations for the authors):*

– Figure 2A has 2 Ic populations but lacks Ia?

– Figure 1c Y axis is shown as "expression level", this should be rather "accessibility level"?

– The Fucci system incorporates genetically encoded Cherry and Venus probes that highlight G1 and S/G2/M phases of the cell cycle in animal cells. Did the authors control for the transgene inactivation in these mice as in some cases the expression of the transgenes may change?

*Reviewer #3 (Recommendations for the authors):*

Related to point (1): projection of the R1A-E scRNAseq clusters onto the scATAC-seq UMPA would be enlightening.

---

## [Author Response]

Essential revisions:1. The scATACseq experiment included a relatively low number (n=2) of mice. Understandably this is a complicated experiment, however, the variation between individual animals remains unknown and may influence the interpretation of the results. The authors are advised to show the clustering analysis for the two animals separately in a supplement to confirm that the changes are seen in both animals, not just in one.

Unfortunately, the two mice are not individually labeled, so the data cannot be separated. Therefore, to address the reviewer's concerns, we have performed another scATAC-seq analysis of TEC in mice of the same age (page 4, line 33). As shown in Figure 1—figure supplement 2 and 3, the data were reproducible, confirming clear and similar separation of proliferating TEC clusters.

Similarly, the individual scRNA-seq UMAPs of the three animals should be included in the supplement. The authors should discuss the limitations of small groups in single-cell experiments in Discussion. Technically the integration of scATAC and scRNA seq results may be feasible.

As with the scATAC-seq, UMAP and clustering analysis from the two experiments of scRNA-seq were displayed separately. These biological duplicates were similar, and consistent conclusions were reached. This finding is described on page 5, line 24, and the data are exhibited in Figure 2_supplement_2. Also, a paragraph has been added to the Discussion (page 14, line 19) describing limitations due to the small number of single-cell analyses.

2. Genes expressed in clusters should be provided in the supplement. Without this information, it is difficult to see the feasibility of the cluster annotations and to compare them with other similar studies. The authors should give an unbiased list of all genes expressed in mTECs, in particular those genes that are expressed in TACs.

We added a heatmap figure of the top 5 genes specifically expressed in each cluster in Figure_2_supplement_1. In addition, the list of genes differentially expressed in each cluster was summarized in Supplementary Table 1.

To harmonize the findings in the field, it would be useful for the readers if the authors would compare their cluster-specific genes for the overlap with TEC single-cell clusters from other studies (Bornstein et al., 2018, Dhalla et al., 2020, Baran-Gale et al., 2020, Wells et al., 2020). In addition, Wells et al., 2020 reported mTEC TACs to be mTEClo preceding Aire expression and giving rise to both Aire+ and Ccl21a+ cells. How do the authors reconcile their results (TAC as Aire+) with Wells et al., paper (TACs as Aire-)? How do they interpret the finding that Wells et al., find Ki67 significantly higher in mTEClo than in mTEChi, implying that cell division precedes high the expression of Aire?

Wells and we reported a cell cluster of TECs expressing high levels of cell proliferation markers in droplet scRNA-seq analysis. It is important to link cell subsets defined by scRNAseq with cell populations defined by flow cytometric analysis.

Wells et al., proposed that Ki67 is a marker of TA-TECs for flow cytometric analysis. Although Ki67^hi^ mTECs were assigned as a part of the Aire-negative MHC class II^lo^ TEC subset, actual data showed that the expression level of MHC class II appears close to the border separating MHCII^lo^ and MHC^hi^ subsets (Figure 2c in Wells et al., *eLife* 2020; 9: e60188), which was also mentioned by the authors. Moreover, the expression level of the Aire protein in Ki67^+^ TECs was broad (Figure 2d in Wells et al., *eLife* 2020; 9: e60188), suggesting that Ki67^+^ TECs may contain both Aire+ and Aire- mTEC subsets. In contrast, we used Fucci/Aire-GFP dual reporter mice to characterize proliferating mTECs that are mCherry^lo^ and Venus^+^. CD80 expression of TA-TECs was high, but appears to be close to the border line separating CD80hi and CD80lo subsets. So, expression level of MHC II, CD80, and Aire of proliferating TECs may be slightly lower and broader, compared to mature mTECs. In flow cytometric analysis, definition of cell types based on cell surface markers depends on how the border lines are set. This may be one reason for the apparent inconsistency between their TA-TECs (MHCll^lo^Aire-) and our TA-TEC (CD80^hi^Aire^+^).

Moreover, we noticed that some Venus^+^ mCherry^lo^ cells (dividing cells in Fucci) are present in the CD80^lo^ fraction, albeit at low frequency (Figure 5—figure supplement 1). This may be consistent with the subcluster analysis in Figure 4, which suggests that the proliferating TEC subset was divided into two; Aire+ subsets (86.2%) and Aire- subsets (13.8%). These data may also support a practical consistency between Wells et al.,’s and our FACS analyses.

It should be noted that not all proliferating cells are transit-amplifying (TA) cells. TA cells are defined as short-lived proliferating cells that link stem cells and mature cell types. Cell fate mapping of possible candidates should be critical for identifying them as “TA”-TEC. Thus, it is important to test the differentiation potential of the proliferating CD80^lo^ fraction into Aire+ and post-Aire mature mTECs in the future.

These points were briefly described in the Discussion (page 13, line 15)

4. The candidate cell population for TACs, cluster R1 expresses Aire and the proliferating cell marker Mki67. R1 also expresses Ccl21a. The authors did subclustering of R1 and found that R1A-D express Aire whereas R1E has a higher expression of Ccl21a. The authors note that "Thus, it is possible that TECs expressing cell-cycle-related genes, proposed by scRNA-seq analysis, contain at least two proliferating TECs subsets having different chromatin accessibilities and gene expression profiles." To confirm that both R1A-D and R1E subsets are proliferating TACs, the authors might show the proliferating gene markers in these subsets. Was Mki67 expressed among all R1 subpopulations? Would this argue for the presence of TAC among both Aire+ and Aire- cell populations? Assuming that TACs as a proliferating cell type should express multiple genes associated with cell cycling, the authors focus on Mki67 only, but did R1 TAC express other proliferating cell markers which would support the claim that these are indeed actively dividing cells?

Thank you for this suggestion. To address these questions, we did sub-cluster the proliferating TEC cluster with a higher resolution (see Materials and methods). Interestingly, the proliferating TEC cluster was divided into 7 subclusters, depending on expression of Aire, Ccl21a, and cell cycle scoring. So, Aire^+^ clusters and Ccl21a^+^ clusters were separated, and these two clusters were further separated by expression of cell cycle phase markers (cell cycle scoring) including Mki67. Thus, these data are consistent with the idea that there may be Aire^+^ and Ccl21a^+^ TECs in this proliferating TEC cluster. We think these findings are important, and data are shown in new Figure 4 and described on Page 8, line 17.

6. The authors focus their study on the CD80high cycling cells (Aire+). Figure 4 show transcription profiles of the isolated cycling CD80+ mTECs. Dots in the "TSA genes" panel (right) don't appear in the "All genes" panel (left). Are those lost dots of TSA genes?

The axis scales were different between the two panels, which might cause confusion. We corrected the axis. Indeed, dots among the “TSA genes” are present in the “All genes” panels.

In Figure 4E (left) low expressed genes seem to be skewed towards Venus- mTEChi (in comparison to high expressed genes). A statistical assessment for the comparisons of the TSA, Aire-dep TSA and Aire-indep TSA profiles to the general profile (Figure 4E and 4G), considering expression levels, would confirm the visual assessment.

To confirm the data in Figure 4E and G (new Figure 5E and G), GSEA analysis was performed and exhibited in Figure 5_supplement_3. Whereas both Aire dep TSA and Aire-indep TSAs were significantly enriched in Venus- mTEChi, the enrichment (enrichment score) of Aire-dep TSAs was higher than that of Aire-indep TSAs.

They should also discuss the reason why m-cherry low cells express a lower level of tissue-specific antigens even though they express Aire? Is Aire expression alone insufficient for TSA expression? Does the transcriptome data provide any mechanistic insight?

As pointed out by this reviewer, it is possible that AIRE expression is not sufficient for inducing TSAs in mTECs. In another scenario, there may be factors that inhibit the AIRE function during cell division in TA-TECs. This point is described in the Discussion (Page 13, line 24). Unfortunately, we have not gained any useful insights about mechanisms from transcriptomic data. Further comparative analysis (such as epigenetic and chromatin structure analyses) between TSA^+^ and TSA^lo^ Aire^+^ TECs will be important to address this issue.

In Fig4F, Aire expression is similar between Venus+ and Venus-. Is it compatible with Figure 4A showing less Aire-expressing cells in Venus+ than in Venus-?

As pointed out by the Reviewer, the fluorescence intensity of Aire GFP in Venus+ mTEChi cells may be lower than that of that of Venus– mTEChi cells in Figure 4A (Figure 5B in revised manuscript). However, as described on Page 9 line 15, a direct comparison between two intensities may be difficult because the fluorescence intensity of Aire GFP in Venus+ mTEChi cells showed a broad peak due to the compensation between GFP and Venus proteins, which have very close fluorescence spectra. In addition to this technical issue, because mTEC^hi^ contains both Aire-positive and Aire-negative cells (Post-Aire mTECs), the average expression level of *Aire* in mTEC^hi^ may have become similar to that of Venus+ cells, which contain mainly Aire-positive cells, in the bulk RNA-seq analysis (Figure 5F)

7. In Figure 6, they showed that RTOC culture of mCherry-low cells produced mCherry-high cells, thereby they claimed that mCherry-low cells differentiated into mCherry-high cells. To substantiate this notion, they should rule out the possibility of selective cell survival of possibly contaminated mCherry-high cells or that transferring adult mTECs into the embryonic cell environment may trigger other signaling pathways that induce the upregulation of the cell cycle and mCherry expression? For example, what about the expression profiles of cell-death/apoptosis-related genes in mCherry low and high cells? What about the cell number of mCherry high cells after RTOC culture using mCherry-high cells alone as a control group? Is the cellularity of survivors comparable to RTOC culture using mCherry-low as shown in figure 6A?

This reviewer noted that mCherry^hi^ contamination of sorted cells could preferentially survive in RTOC. We compared expression of pro-apoptotic genes between mCherry^hi^ and mCherry^lo^. GSEA analysis suggested that pro-apoptotic genes were rather higher in mCherry^hi^ than mCherry^lo^ (Figure_7_figure_supplement_1D). Thus, it is unlikely that mCherry^hi^ could be more resistant to apoptosis, thereby surviving preferentially in RTOC.

Moreover, we have done RTOC using only mCherry^hi^ cells. A purity check of sorted mCherry^lo^ cells indicated that the contamination was below 0.05% (Figure_7_figure_supplement_1C). Therefore, the number of mCherry^hi^ cells mixed with RTOC was changed from 200, 500, and 1000 cells, which would be equivalent to the assumed contamination of approximately 0.2%, 0.5%, and 1 % of mCherry^hi^ cells during cell sorting, respectively. Data suggested that surviving cells are 1, 0, and 5 cells in these experiments. Thus, it is unlikely that detected mCherry^hi^ in RTOC were simply survivors of contaminated mCherry^hi^. In addition, this experiment rules out the possibility that mCherry^hi^ becomes more proliferative in the embryonic thymus environment. These points are briefly described in Page 11, line 1.

Would the same happen if they would transfer these cells to the adult thymus?

This is a good suggestion because a similar experiment was previously carried out to address the potential of mTEC stem cells (Sekai et al., J. Immunol. Methods 467, 29, 2019). In that experiment, they used 5x 10^5^ cultured cells for the adult thymic transfer. Because we cannot culture TA-TECs to date, to prepare this number of cells from mice, we would need to use 20 thymi per experiment and would then have to sort the TA-TECs. This is quite a difficult experiment in reality. However, our preliminary trial using 1 x 10^5^ TA-TECs was not successful in that we failed to detect transferred cells. Together with the low efficiency of RTOC and the failure of MHC mismatched RTOC, we think some kind of niches are necessary to maintain TA-TECs. Determination of such niches will be an important study in the future.

8. The authors nicely confirm, by the fine analysis of their scRNA-seq data, that the TAC population contains Aire+ cells and Ccl21+ cells in a visually mutually exclusive manner. However, they don't formally clarify whether the Ccl21-expressing TACs have differences in their chromatin accessibility pattern compared to the Aire-expressing TACs.

Integration of TAC-subcluster analysis with scATAC-seq revealed that chromatin accessibility of Aire^+^ proliferating TECs and Ccl21a^+^ proliferating cells may not be very different. This was briefly mentioned on Page 8 line 26. However, this point should be carefully concluded because a relatively low number cells of each subcluster was analyzed. So, it is important to compare chromatin structure between these clusters after isolation using a flow cytometer.

It's also worth showing the expression of CD80 in the scRNA-seq UMAP of TACs alone and in the one of all TECs (Figure 2). This would notably allow to determine whether CD80 expression is restricted to Aire-positive TACs or encompasses Aire-negative TACs (Ccl21+).

We demonstrated CD80 expression of the proliferating TEC subcluster in a Feature plot (Figure 4D). Whereas CD80 positive cells are fewer, it is likely that CD80 expressing cells are present among Aire-positive cells, but not among Ccl21+ cells in the proliferating TEC clusters.

Also, projection of the R1A-E scRNAseq clusters onto the scATAC-seq UMPA would be enlightening.

As mentioned above, we projected the proliferating TEC subclusters (R1A-G) onto the scATAC-data. The analysis did not show clear evidence of a difference in chromatin accessibility among these clusters. The result is briefly described in the main text (Page 8, line 26), and the data are presented in Figure 4-figure_supplement_1.

9. Trajectory analyses provide a nice confirmation of published results identifying a trajectory from TACs to mTEChi. However, the authors don't discuss whether their data support a potential trajectory from TACs to mTEClo (already suggested/ reported) which seems to be present in Figure 3-Figure sup 2B. Would this mean that some TACs could mature into Ccl21+ mTECs (mTEClo)? and if so, how Aire and Ccl21 are expressed in these TACs? Which TAC sub-cluster do they belong to?

This point may be related to the question of whether the proliferating TEC subset in scRNA-seq is a single population. We think that velocity analysis of our scRNA-seq data did not show a clear trajectory from the proliferating TEC cluster to Ccl21+ mTECs. However, since the Ccl21a+ cluster is present in the proliferating TEC cluster (R1), it may be possible that these cells differentiate into Ccl21+ mTECs (mTEClo). To date, trajectory analysis of the proliferating Ccl21a cluster has been difficult because of the low cellularity in our scRNA-seq dataset.

10. In figure S7, the data of fetal thymi showed that proliferating fetal mTECs might have a gene expression profile different from the adult counterpart. One caveat of the interpretation of the data is that the difference between the public data and the authors' data might be attributed to a batch effect because the clusters from two data sets seemed almost completely discrete. They should mention how they processed the data to diminish the risk of such artifacts. Or, it is better to add the data of fetal thymi obtained by the authors themselves, if possible.

In Figure S7, a batch effect was removed using the Seurat function, in which selected “anchor” genes commonly expressed in cells are used to remove the batch effect. This function was used to remove the batch effect between our data and those of others, so it turned out to be effective. We also used the Harmony algorithm to remove the batch effect (Figure 8—figure supplement 1C). Again, TEC data of fetal thymus differ from adult TECs. We further integrated fetal thymus data with previously reported scRNA-seq data of TECs (Bornstein et al., Nature 2018; 559:622, Wells et al., *eLife* 2020;9:e60188, Dhalla et al., *eLife*. 2020 9:e56221). Data analysis suggested that TECs of fetal thymus seem to be different from adult TEC clusters (Figure 8—figure supplement 2). Overall, it is likely that fetal TECs are considerably different from adult TECs in terms of their gene expression profiles. Unfortunately, we did not perform reproducing experiments for fetal TEC scRNA-seq by ourselves, because it is fairly costly and time-consuming to obtain scRNA-seq data of TECs from various embryonic days.